

1 **Biogeochemical constraints on the origin of methane in an alluvial aquifer: evidence**

2 **for the upward migration of methane from a coal seam.**

4 Charlotte P. Iverach[1,2,*], Sabrina Beckmann[3], Dioni I. Cendón[1,2], Mike Manefield[3], Bryce

5 F.J. Kelly[1]

7 [1]Connected Waters Initiative Research Centre, UNSW Australia, UNSW Sydney, NSW,

8 2052, Australia.

9 [2]Australian Nuclear Science and Technology Organisation, New Illawarra Rd, Lucas

10 Heights, NSW, 2234, Australia.

11 [3]School of Biotechnology and Biomolecular Sciences, UNSW Australia, UNSW Sydney,

12 NSW, 2052, Australia.



**Geochemical and microbiological indicators of methane (CH$_4$) production, oxidation**
**and migration processes in groundwater are important to understand when**
**attributing sources of gas. The processes controlling the natural occurrence of CH$_4$**
**in groundwater must be understood, especially when considering the potential**
**impacts of the global expansion of coal seam gas production on groundwater quality**
**and quantity. We use geochemical and microbiological data, along with**
**measurements of CH$_4$ isotopic composition ($\delta^{13}$C-CH$_4$), to determine the processes**
**acting upon CH$_4$ in a freshwater alluvial aquifer that directly overlies coal measures**
**targeted for coal seam gas production in Australia. Microbial and geochemical data**
**indicate that there is biogenic CH$_4$ in the aquifer, but no methanogenic microbial**
**activity. In addition, microbial community analysis showed that aerobic oxidation of**
**CH$_4$ is occurring. The combination of microbiological and geochemical indicators**
**suggests that the most likely source of CH$_4$, where it was present in the freshwater**
**aquifer, is the upward migration of CH$_4$ from the underlying coal measures.**

**Keywords:** Methane migration, groundwater, biogeochemistry, methanogenesis,
methanotrophy, coal seam gas, aquifer connectivity

**1    Introduction**
Interest in methane (CH$_4$) production and degradation processes in groundwater is driven
by the global expansion of unconventional gas production. There is concern regarding
potential impacts of gas and fluid movement, as well as depressurisation, on groundwater
quality and quantity in adjacent aquifers used to support other industries (Atkins et al.,
2015; Heilweil et al., 2015; Iverach et al., 2015; Moritz et al., 2015; Zhang et al., 2016).





In groundwater, $CH_4$ can originate from numerous sources (Barker and Fritz, 1981).
The two main sources of $CH_4$ in shallow groundwater are biological production
(biogenic) and upward migration of $CH_4$ from deeper geological formations (thermogenic
to mixed thermo-biogenic to biogenic) (Barker and Fritz, 1981; Whiticar, 1999). This
upward migration is via natural pathways such as geological faults and fracture networks
(Ward and Kelly, 2007), however it can also be induced via poorly installed wells and
faulty well casings (Barker and Fritz, 1981; Fontenot et al., 2013). The main focus of the
debate about the occurrence of $CH_4$ in groundwater is whether it is naturally occurring or
has been introduced by human activities. This research tests the hypothesis that a
combination of geochemical indicators and microbiological data can inform production,
degradation and migration processes of $CH_4$ in the Condamine River Alluvial Aquifer
(CRAA) in Australia. This freshwater aquifer directly overlies the Walloon Coal
Measures (WCM), the target coal measures for coal seam gas (CSG) production in the
study area. Thus, our study has ramifications for global unconventional gas studies that
investigate connectivity issues to freshwater aquifers.
Methane is subject to many production and degradation processes in groundwater
(Whiticar, 1999). The carbon isotopic composition of $CH_4$ ($\delta^{13}C$-$CH_4$) gives insight into
the source (Quay et al., 1999), but oxidation processes may enrich or deplete this
signature (Yoshinaga et al., 2014). Therefore, it is very difficult to determine the potential
source of $CH_4$ and processes occurring using $CH_4$ concentration and isotopic data alone.
Previous studies have used geochemical indicators, such as the concentration of
sulfate [$SO_4^{2-}$], nitrate [$NO_3^-$] and nitrite [$NO_2^-$], and the carbon isotopic composition of
dissolved inorganic carbon ($\delta^{13}C$-DIC) and dissolved organic carbon ($\delta^{13}C$-DOC) to
attribute the source of $CH_4$ in groundwater (Valentine and Reeburgh, 2000; Kotelnikova,
2002; Antler, 2014; Green-Saxena et al., 2014; Antler et al., 2015; Hu et al., 2015;


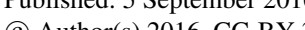

Segarra et al., 2015; Sela-Adler et al., 2015; Currell et al., 2016). Other studies have
shown that the presence of active methanogenesis can be determined using isotopes of
hydrogen in the $CH_4$ ($\delta^2H$-$CH_4$), and the surrounding formation water ($\delta^2H$-$H_2O$)
(Schoell, 1980; Whiticar and Faber, 1986; Whiticar, 1999; Currell et al., 2016).
Additionally, recent studies have used clumped isotopes of $CH_4$ and their temperature
interpretations to ascribe a thermogenic versus biogenic source in groundwater (Stolper et
al., 2014). However, non-equilibrium (kinetic) processes may be responsible for an
overestimation of $CH_4$ formation temperatures (Wang et al., 2015). Therefore, combining
geochemistry and microbiology provides a robust method to assess $CH_4$ origin, as it
directly discriminates between microbiological communities involved in either production
or degradation processes.

Throughout the world the occurrence of freshwater aquifers adjacent to

unconventional gas production is common (Osborn et al., 2011; Moore, 2012; Roy and
Ryan, 2013; Vidic et al., 2013; Vengosh et al., 2014; Moritz et al., 2015). We have
previously shown that there may be local natural connectivity between the WCM and the
CRAA (Iverach et al., 2015). Here we show that a combination of geochemical data
([$CH_4$], [$SO_4^{2-}$], [$NO_3^-$], [$NO_2^-$], $\delta^{13}C$-$CH_4$, $\delta^{13}C$-DIC, $\delta^{13}C$-DOC and $\delta^2H$-$H_2O$), as well as
characterisation of microbiological communities present, can inform the discussion
surrounding the occurrence of $CH_4$, and its potential for upward migration in the
groundwater of the CRAA.

**1.1   Geochemical indicators of methanogenic processes**
Methanogenesis via acetate fermentation (Eq. 1) and carbonate reduction (Eq. 2) can be
restricted in groundwater with abundant dissolved $SO_4^{2-}$ (> 19 mg/L) (Whiticar, 1999),



because sulfate reducing bacteria (SRB) can outcompete methanogenic archaea for
reducing equivalents (Lovley et al., 1985).

$CH_3COOH \rightarrow CH_4 + CO_2$ (1)

$CO_2 + 8H^+ + 8e^- \rightarrow CH_4 + 2H_2O$ (2)

Therefore, the presence or absence of $[CH_4]$ and $[SO_4^{2-}]$ are good preliminary indicators
of the potential for methanogenesis.

In addition, the $\delta^{13}C$-$CH_4$ of the underlying WCM in the study area has been

characterised (Papendick et al., 2011; Hamilton et al., 2012; Hamilton et al., 2014). Thus
the isotopic signature can be used to identify the potential source of the $CH_4$, however
localised formation and oxidation processes that may occur either in the aquifer or during
transport can confound the interpretation of mixing versus oxidation processes.

The isotopic composition of DIC and DOC are also useful indicators of $CH_4$

processes, as they can be used to determine the occurrence of methanogenesis
(Kotelnikova, 2002; Wimmer et al., 2013). Kotelnikova (2002) found that $^{13}C$-depletion
of $\delta^{13}C$-DOC in combination with a $^{13}C$-enrichment of $\delta^{13}C$-DIC was characteristic of
methanogenesis in groundwater, consistent with the reduction of $^{12}CO_2$ by autotrophic
methanogens. Conversely, $\delta^{13}C$-DIC data are useful because DIC produced during $CH_4$
oxidation was found to have a characteristically $^{13}C$-depleted signature (as depleted as -
50‰) (Yoshinaga et al., 2014; Hu et al., 2015; Segarra et al., 2015).

**1.2    Methane oxidation in freshwater**
In groundwater, $CH_4$ is oxidised by methane-oxidising bacteria (MOB; methanotrophs)
that can utilise $CH_4$ as their sole carbon and energy source. These methanotrophs are
grouped within the *Alpha-* and *Gamma*-Proteobacteria (comprising type I and type II
methanotrophs) and the Verrucomicrobia (Hanson and Hanson, 1996). The first step of





aerobic $CH_4$ oxidation is the conversion of $CH_4$ to methanol. This is catalysed by the
particulate $CH_4$ monooxygenase (*pMMO*) encoded by the *pmoA* gene, which is highly
conserved and used as a functional marker (Hakemian and Rosenzweig, 2007; McDonald
et al., 2008). All known methanotrophs contain the *pmoA* gene, with members of
*Methylocella* the exception (Dedysh et al., 2000; Dunfield et al., 2003). Type II
methanotrophs and some type I members of the genus *Methylococcus* contain the *mmoX*
gene, which encodes a soluble $CH_4$ monooxygenase (McDonald et al., 1995; Murrell et
al., 2000). Recently, new groups of aerobic and anaerobic MOB distantly related to
known methanotrophic groups have been discovered (Raghoebarsing et al., 2006;
Stoecker et al., 2006; Op den Camp et al., 2009). Geochemically, aerobic $CH_4$ oxidation
has been previously coupled to denitrification in groundwater (Zhu et al., 2016).

Besides methanotrophic bacteria, anaerobic $CH_4$ oxidising archaea (ANME) also

play a significant role in the oxidation of $CH_4$ in both freshwater and saline water sources
(Knittel and Boetius, 2009). These anaerobic methanotrophs are associated with the
methanogenic Euryarchaeota within the clusters ANME-1, ANME-2, and ANME-3 and
are closely related to the orders *Methanosarcinales* and *Methanomicrobiales* (Knittel et
al., 2003; Knittel et al., 2005). Geochemical indicators can provide evidence for the
occurrence of AOM, such as the prevalence of certain electron acceptors ($SO_4^{2-}$, $NO_3^-$,
$NO_2^-$ and $Fe^{2+}$) (Valentine and Reeburgh, 2000; Ettwig et al., 2010; Sivan et al., 2011;
Antler, 2014; Green-Saxena et al., 2014) and denitrification processes occurring in the
groundwater (Ettwig et al., 2008; Nordi and Thamdrup, 2014; Timmers et al., 2015).

**2    Study Area**
The CRAA is the primary aquifer in the Condamine Catchment (Figure 1). It is used
primarily for irrigated agriculture, stock and domestic water supplies. There has been



increased interest in the presence of CH$_4$ in the aquifer due to expanding CSG production
to the north-west of the study area (Figure 1). CSG production began in 2006 (Arrow
Energy, 2015) and has been expanding in the decade since then. This has raised concerns
regarding the quality and quantity of the groundwater in the CRAA.

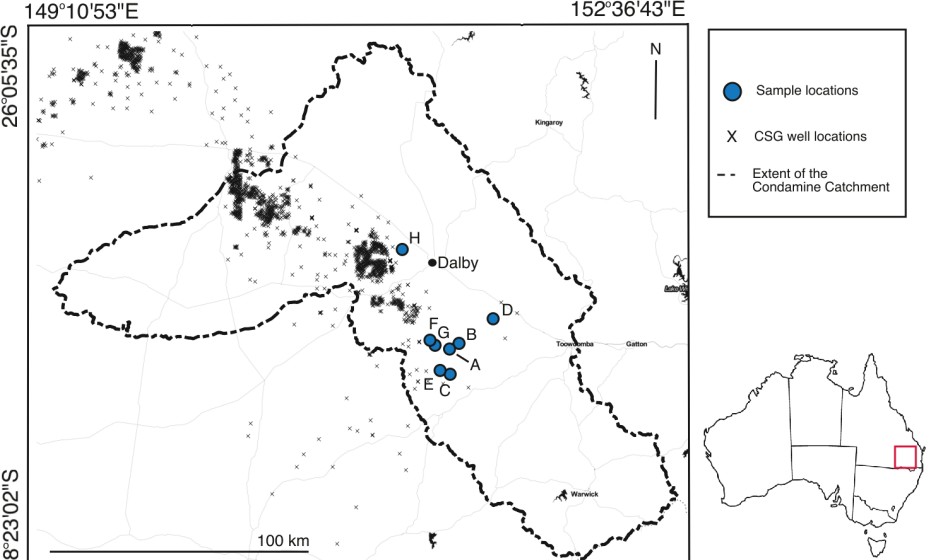


**Figure 1.** Site map showing the extent of the study area and sample locations within the Condamine
Catchment, south-east Queensland, Australia. Map created in QGIS; data and imagery: Statem Toner, Open
Street Map and contributors, CC-BY-SA (QGIS, 2015). Modified with Corel Painter 2015 (Corel
Corporation, 2015).

**2.1    Hydrogeological setting**
The CRAA sits within the Surat Basin, which sits within the Great Artesian Basin (GAB)
in south-east Qld, Australia (Figure 1). Aquifers in the GAB vary between semi-confined
and confined (Kelly and Merrick, 2007; Dafny and Silburn, 2014).



The environment of deposition for the Surat Basin was fluvio-lacustrine in the late

Triassic-Jurassic and shallow marine and coastal in the Cretaceous (Hamilton et al.,

2012). The middle-Jurassic WCM are a group of low-rank coal seams in the Surat Basin

targeted for CSG production (Hamilton et al., 2012). The WCM are thicker (150 m to 350

167       m) along the western margin of the CRAA and thin to around 50 m in the east, where the

unit outcrops (KCB, 2011), however, only around 10 % of this is coal. The unit consists

of very fine- to medium-grained sandstone, siltstone, mudstone and coal, with minor

calcareous sandstone, impure limestone and ironstone (KCB, 2011). The coal consists of

numerous discontinuous thin lenses separated by sediments of low permeability (Hillier,

2010). The unit dips gently to the west (around 4º), which is consistent with the general

trend of the Surat Basin in this region.

The WCM overlie the Eurombah Formation (consisting of conglomerate sandstone

with minor siltstones and mudstone beds) and underlie the Kumbarilla Beds (mainly

sandstone, with lesser mudstone, siltstones and conglomerates) (KCB, 2011).

The unconfined CRAA fills a paleovalley that was carved through the GAB

(including the WCM). The valley-filling sediments are composed of gravels and fine- to

course-grained channel sands interbedded with floodplain clays and, on the margins,

colluvial deposits, which were deposited from the mid-Miocene to the present (Kelly and

Merrick, 2007; Kelly et al., 2014). The valley-filling sediments have a maximum

thickness of 134 m near Dalby (Dafny and Silburn, 2014). Along the eastern margin of

the valley, the CRAA is bounded by the Main Range Volcanics. Estimations of the

sources and quantity of recharge to the CRAA vary widely, however streambed recharge

is generally considered to be the major source of freshwater to the aquifer (Dafny and

Silburn, 2014).





A low permeability layer (ranging from 8 x $10^{-6}$ to 1.5 x $10^{-1}$ m/d) has been reported
between the CRAA and the underlying WCM (KCB, 2011; QWC, 2012). This has been
referred to as the 'transition layer' (QWC, 2012) or a 'hydraulic basement' to the
alluvium (KCB, 2011). However, the thickness of this layer varies between 30 m thick in
some areas to completely absent in others. Thus, in some places the WCM immediately
underlies the CRAA (Dafny and Silburn, 2014). This suggests that there is some level of
connectivity between the CRAA and the WCM. Huxley (1982) and Hillier (2010) both
suggest that the general decline in water quality downstream is due to some net flow of
the more saline WCM water into the CRAA. Connectivity between the formations is not
well understood; however, studies have been conducted to better understand the
movement of both water and gas between the two aquifers. Duvert et al. (2015) and Owen
and Cox (2015) both used hydrogeochemical analyses to show that there was limited
movement of water between the two formations. However, Iverach et al. (2015) used the
isotopic signature of $CH_4$ in the groundwater to show that there was localised movement
of gas between the coal measures and the overlying aquifer. This research provides
additional insight to inform the debate about the degree of connectivity between the
WCM and the CRAA. The microbiological insights also inform the global research on
biological $CH_4$ production and degradation in alluvial aquifers, in particular for zones
distal to the river corridor.

**3    Methods**
From 22 January 2014 to 31 January 2014 we collected groundwater samples for
geochemical analysis from 8 private irrigation boreholes in the Condamine Catchment.
Iverach et al. (2015) outlines the complete methods for sample collection for [$CH_4$] and





$\delta^{13}C\text{-}CH_4$ and subsequent analysis. The 8 samples collected from the unconfined CRAA
are representative of the aquifer, given their varied depths and locations.
Groundwater samples were collected by installing a sampling tube 2 m inside the
pump outlet of the borehole to avoid the air-water interface at the sampling point. Field
parameters (electrical conductivity (EC), oxidation-reduction potential (ORP), dissolved
oxygen (DO), temperature (T) and pH) were monitored in a flow cell to ensure
stabilisation before samples were collected. The boreholes had been pumping
continuously over the preceding month for irrigation and so stabilisation of the field
parameters was reached within minutes. Groundwater samples for major anions and
water-stable isotopes ($\delta^2H\text{-}H_2O$ and $\delta^{18}O\text{-}H_2O$) were collected after passing the water
through a 0.45 μm, high-volume groundwater filter, which was connected to the pump
outlet. Groundwater for anions and water stable-isotopes were stored in 125 mL high-
density polyethylene (HDPE) bottles and 30 mL HDPE bottles, respectively. Both had no
further treatment. The water for $\delta^{13}C\text{-}DIC$ and $\delta^{13}C\text{-}DOC$ was further filtered through a
0.22 μm filter and stored in 12 mL Exetainer vials and 60 mL HDPE bottles, respectively.
Samples to be analysed for DIC were refrigerated at 4 ºC and samples to be analysed for
DOC were frozen within 12 hours of collection.
Groundwater samples for the microbiological analyses were collected between 8
December 2014 to 11 December 2014, and were collected from the same 8 private
irrigation boreholes used for the geochemical analyses. Groundwater samples for
microbiological analysis were collected in 2 L Duran Schott bottles and sealed (gas tight).
The groundwater was filtered through a 0.2 μm filter (Merck Millipore). We use aspects
of the geochemical data collected in the January campaign to inform our interpretation of
the microbial results from the December campaign.



### 3.1 Geochemical analyses

The major ion chemistry in the groundwater samples was analysed at the Australian Nuclear Science and Technology Organisation (ANSTO) using Inductively Coupled Plasma - Ion Chromatography for anions. The samples for $\delta^2$H-$H_2O$ and $\delta^{18}$O-$H_2O$ were analysed at ANSTO and are reported as ‰ deviations from the international standard V-SMOW (Vienna Standard Mean Ocean Water). $\delta^{18}$O samples were run using an established equilibration, continuous flow IRMS method and $\delta^2$H samples were run using an on-line combustion, dual-inlet IRMS method.

The isotopes of carbon in DIC were analysed at ANSTO using an established method on a Delta V Advantage mass spectrometer, and a GasBench II peripheral. The results are reported as a ‰ deviation from IAEA secondary standards that have been certified relative to V-PDB for carbon. The isotopes of carbon in DOC were analysed at UC-Davis Stable Isotope Facility and results are reported as ‰ and are corrected based on laboratory standards calibrated against NIST Standard Reference Materials with an analytical precision of ± 0.6‰. Samples were run using a total organic carbon (TOC) analyser connected to a PDZ Europa 20-20 IRMS using a GD-100 Gas Trap interface. The [$SO_4^{2-}$] were too low in 6 of the 8 samples for $\delta^{34}$S and $\delta^{18}$O analysis. The remaining 2 samples were analysed for their sulfur and oxygen isotope compositions at the University of Calgary Isotope Science Laboratory. Sulfur isotope ratios were analysed using a Continuous Flow-Isotope Ratio Mass Spectrometry (CF-EA-IRMS) with an elemental analyser interfaced to a VG PRISM II mass spectrometer. The results are reported against V-CDT (Vienna Cañon Diablo Troilite). The oxygen isotope ratio was determined using a high temperature reactor coupled to an isotope ratio mass spectrometer in continuous flow mode.



### 3.2 DNA extraction and Illumina sequencing


DNA was extracted from the biomass collected from filtering 2 L of groundwater. Briefly,
DNA was extracted using a phenol-chloroform extraction method as described by Lueders
et al. (2004). Subsequently, the DNA was precipitated using polyethylene glycol 6000
(Sigma Aldrich), and the DNA pellet was washed using 70 % (v/v) ethanol and
resuspended in 50 μL nuclease free water (Qiagen). DNA concentration and purity were
determined by standard agarose gel electrophoresis and fluorometrically using RiboGreen
(Qubit Assay Kit, Invitrogen) according to the manufacturer's instructions. The extracted
DNA was used as a target for Illumina sequencing. Amplicon libraries were generated by
following Illumina's 16S Metagenomic Sequencing Library Preparation Protocol, using
12.5 ng of template DNA per reaction. The number of cycles for the initial PCR was
reduced to 21 to avoid biases from over-amplification. The following universal primer
pair was used for the initial amplification, consisting of an Illumina-specific overhang
sequence and a locus-specific sequence:
926F_Illum(5'-
TCGTCGGCAGCGTCAGATGTGTATAAGAGACAG[AAACTYAAAKGAATTGRC
CG]-3'),
1392R_Illum(5'-
GTCTCGTGGGCTCGGAGATGTGTATAAGAGACAG[ACGGGCGGTGTGTRC]-3').
This universal primer pair targets the V6-V8 hyper-variable regions of the 16S ribosomal
RNA gene and has been shown to capture the microbial diversity of Bacteria and Archaea
in a single reaction (Wilkins et al., 2013). PCR products were purified using a magnetic
bead capture kit from Agencourt AMPure XP beads (Beckman Coulter) and quantified
using a fluorometric kit (RiboGreen, Qubit Assay Kit, Invitrogen). Purified amplicons

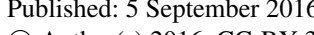



were subjected to the Index PCR using the MiSeq platform (Ramaciotti Centre for
Genomics, UNSW Australia) according to the manufacturer's specifications. Illumina
sequences were checked for quality (FastQC, BaseSpace) and analysed using the
BaseSpace cloud computing platform (Illumina, 2016) and MOTHUR (Schloss, 2009)
with modified protocols (Schloss et al., 2009; Kozich et al., 2013). Taxonomy was
assigned against the SILVA Database (Silva, 2016). To ensure even sampling depth for
subsequent analyses, OTU abundance data were rarefied to the lowest number of
sequences for a sample (8,300 sequences per sample).

**3.3   Quantification of bacterial and archaeal 16S rRNA and functional genes**
Quantitative real-time PCR was used to determine abundances of bacterial and archaeal
16S rRNA gene targets and functional gene targets (*mcrA, pmoA, mmoX,* and *dsrA*), using
the MJ Mini™ 96 Well Thermal Cycler (Bio-Rad, Hercules, CA). Each qPCR 25 µL
reaction mixture contained 12.5 µL of premix solution from an iQ SYBRGreen qPCR Kit
(Bio-Rad), 8 µL PCR-grade water, 1.5 µL of each primer (final concentration 0.2 – 0.5
µM), and 2 µL of template DNA (10 ng). Bacterial and archaeal 16S rRNA genes were
amplified using the primer pairs 519F/907R (Lane 1991; Muyzer et al., 1995) and
SDArch0025F/SDArch0344R (Vetriani et al., 1999). *McrA* and *dsrA* sequence fragments
were amplified using the primer pairs ME1F/ME3R (Hales et al., 1996) and 1F/500R
(Wagner et al., 1998; Dhillon et al., 2003). QPCRs were performed as described
previously by Wilms et al. (2007). *PmoA* qPCR was performed using the *pmoA* primer-
pair A189F (Holmes et al., 1999) and mb661R (Kolb et al., 2003) with a final total
concentration of 0.8 µM. The qPCR programme for the amplification was performed as
follows: 95ºC for 3 min followed by 40 cycles of 95ºC for 30 s, 64ºC for 45 s and 68ºC
for 45 s. The *mmoX* gene fragment was quantified using the prime pairs mmoX-ms-945f



and mmoXB-1401b at a final concentration of 0.8 μM. The qPCR conditions for the
*mmoX* was as follows: 94ºC for 3 min followed by 45 cycles of 94ºC for 1 min, 50ºC for
1 min and 72ºC for 1 min. Bacterial and archaeal targets were measured in at least three
different dilutions of DNA extracts (1:10, 1:100, 1:1000) and in triplicate. PCR products
were checked by gel electrophoresis, using 2 % (w/v) agarose with TBE buffer (90 mM
Tris, 90 mM boric acid, 2 mM $Na_2$-EDTA; pH 8.0). The specificity of the reactions was
confirmed by melting curve analysis and agarose gel electrophoresis to identify non-
specific PCR products. Amplification efficiencies for all reactions ranged from 96.3 % to
110.5 % with an $r^2$ value of > 0.99 for standard curve regression. DNA calibration
standards for qPCR were prepared as follows. The *mcrA, dsrA, pmoA,* and *mmoX* genes
were amplified from pure cultures of *Methanosarcina barkeri*[T] (DSM 800), *Desulfovibrio*
*vulgaris*[T] (DSM 644), *Methylosinus sporium*[T] (DSM 17706), and *Methylocella silvestris*[T]
(DSM 15510; DZMZ Germany). The PCR amplicons were purified using the DNA Clean
and Concentrator[TM]-5 kit (Zymo Research, Irvine, CA), and eluted into 20 μL DNA
elution buffer. DNA concentrations were quantified with 2 μL DNA aliquots using the
Qubit® dsDNA BR Assay Kit (Invitrogen, Life Technologies, Carlsbad, CA). Purified
target gene PCR products were cloned into plasmids following the manufacturer's
instructions for the pGEM® – T Easy Vector System (Promega, Madison, WI).

**4   Results and Discussion**
**4.1   Previous $\delta^{13}$C-CH$_4$ investigation**
A previous study by Iverach et al. (2015) analysed the $\delta^{13}$C-CH$_4$ in the groundwater from
an off-gassing port on the 8 private irrigation boreholes studied here (samples A-H)
(Supplementary Table S3 online). These measurements were understood to have been
mixing with regional background atmospheric CH$_4$ (1.774 ppm; -47‰) and therefore

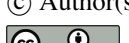


mixing plots were used to infer the isotopic source signature of the $CH_4$ off-gassing from
the groundwater. Iverach et al. (2015) found that samples E, G, and H plotted on a
regression line that had an isotopic source signature of -69.1‰ (90% CI, −73.2‰ to
−65.0‰), indicative of a biological source. However, samples A, B, C, D and F plotted on
a regression line that had an isotopic source signature of -55.9‰ (90% CI, −58.3‰ to
−53.4‰), suggesting either oxidation was occurring at the source or there was potential
upward migration of $CH_4$ from the underlying WCM.

**4.2     Limited geochemical and microbiological potential for methanogenesis in the**
**groundwater**
To further elucidate the source of the $CH_4$ reported in the groundwater (Iverach et al.,
2015), Illumina sequencing and quantitative real-time PCR (qPCR) was used to target
bacterial and archaeal 16S rRNA genes, as well as specific functional genes (*mcrA*, *pmoA*,
*mmoX* and *dsrA*) associated with $CH_4$ metabolism. Microbial abundances estimated by
SYBR Green I counts were between $10^3$ and $10^5$ cells/mL throughout all groundwater
samples (Figure 2). This was congruent with the qPCR data observed for bacterial and
archaeal cell concentrations.





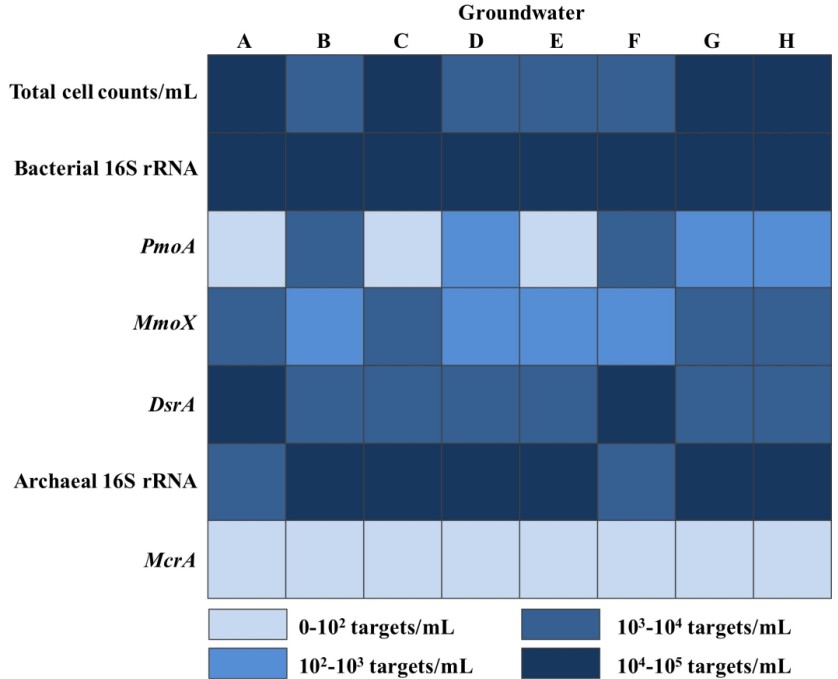

**Figure 2.** Total cell concentration and copy number abundances of bacterial and archaeal 16SrRNA genes and functional key genes for aerobic $CH_4$ oxidation (*pmoA* and *mmoX* genes), $CH_4$ production (*mcrA* gene) and sulfate reduction (*dsrA* gene) in the groundwater carried out by quantitative (q)PCR. Low abundances are highlighted in light blue. High abundances are highlighted in dark blue.

The groundwater community was primarily composed of bacteria (79 to 90 %), whilst archaea made up 10 to 21 % (Figure 3). The bacterial and archaeal community composition did not vary significantly between groundwater samples. Most of the bacterial sequences belonged to the phyla Proteobacteria *(α-δ)*, Acidobacteria, Actinobacteria, Firmicutes and the Bacteroidetes/Chlorobi group (Figure 3). The phylum Thaumarchaeota dominated the archaeal communities with a relative abundance of 81 to 99 %, while Crenarchaeota made up 1 to 3 % of the archaeal community. Further sequences were related to other (if < 1 % relative abundance) and unclassified Bacteria and Archaea. No members of the Euryarchaeota, comprising the methanogenic archaea,



were observed. The archaeal *mcrA* gene, which encodes the methyl coenzyme M
reductase, was not detected in any of the groundwater samples (detection limit < 10
cells/mL; Figure 2). This was consistent with the Illumina sequencing results, and
suggests that the $CH_4$ observed off-gassing from the groundwater was not being produced
locally.

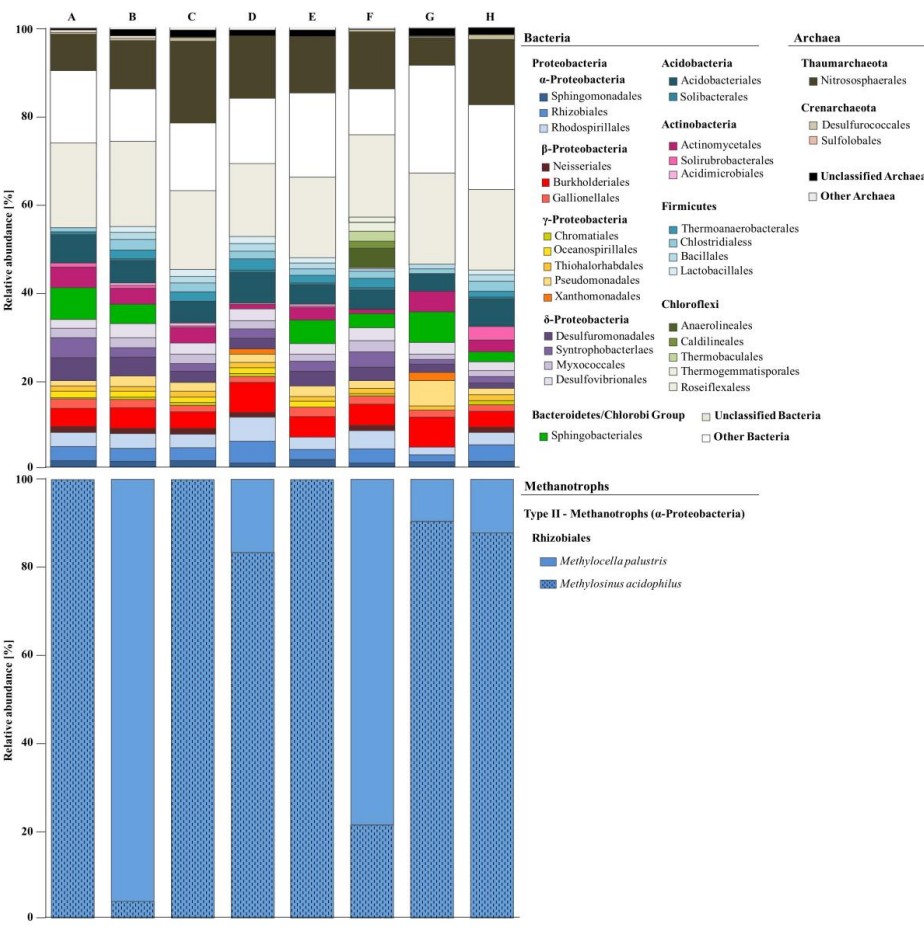

**Figure 3.** Bacterial, archaeal, and methanotrophic community profiles and relative abundances detected by
Illumina sequencing.

Our geochemical data also showed no evidence for the occurrence of
methanogenesis in the groundwater. As previously stated, a $^{13}C$-enrichment in $\delta^{13}C$-DIC





coupled with a $^{13}$C-depletion in the $\delta^{13}$C-DOC is characteristic of methanogenesis
(Kotelnikova, 2002). Our groundwater data showed no correlation between $\delta^{13}$C-DOC
and $\delta^{13}$C-DIC (Figure 4a), and the most $^{13}$C-enriched $\delta^{13}$C-DIC was also the second
highest enriched $\delta^{13}$C-DOC value. Additionally, on a stable water isotope plot (Figure 4b;
Supplementary Table S1 online), it is evident that there is no noticeable $\delta^2$H-enrichment
that can be ascribed to methanogenesis in any of the groundwater samples (Cendón et al.,

2015).

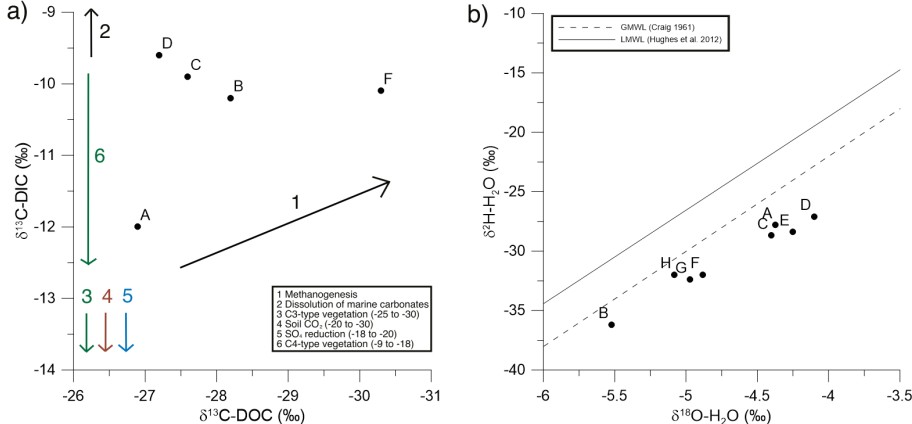


**Figure 4.** (a) A plot of $\delta^{13}$C-DOC vs. $\delta^{13}$C-DIC, highlighting the absence of correlation between these
geochemical data, indicating that there is no methanogenic end member in our samples. Samples E, G and H
are omitted because they were below the detection limit for $\delta^{13}$C-DOC (Supplementary Table S1.). Arrow 1
delineates the expected trend for methanogenesis and arrow 2 is the expected trend for the dissolution of
marine carbonates (Currell et al., 2016). Arrows 3-6 highlight expected ranges for $\delta^{13}$C-DIC that are off the
scale of the graph (Currell et al., 2016). (b) A plot of $\delta^{18}$O-H$_2$O vs. $\delta^2$H-H$_2$O showing that there is no $^2$H-
enrichment in any of the groundwater samples. The GMWL (Craig, 1961) and LMWL (Hughes and
Crawford, 2012) are also displayed.

These geochemical analyses, along with the lack of classified methanogens, suggest
that biogenic CH$_4$ production is not one of the major processes producing CH$_4$ in the
CRAA. Therefore, the CH$_4$ reported in all samples in Iverach et al. (2015) must be



coming from another source. We propose that the upward migration of $CH_4$ from the
WCM must be considered as the potential source. The isotopic signature of $CH_4$ from the
deeper coal measures has been characterised between -58.5‰ and -45.3‰, indicating
thermogenic $CH_4$ with a secondary biogenic component (Papendick et al., 2011; Hamilton
et al., 2012; Hamilton et al., 2014). Five of the 8 samples analysed in this study have an
isotopic source signature within this range, as reported in Iverach et al. (2015). This
implies that upward migration from the deeper WCM is the source of the $CH_4$ detected in
the groundwater.
However, the remaining 3 samples (samples E, G, and H) have a typically biogenic
isotopic source signature (-69.1‰). This could be the result of the replacement of
typically thermogenic gas in the shallow WCM by biogenic gas (Faiz and Hendry, 2006).
Thus, these three sites are potentially sourcing biogenic $CH_4$ from the shallow WCM,
resulting in a biological source signature despite the absence of methanogens in the
overlying aquifer.

**4.3    Sulfate reducers and aerobic methanotrophs potentially outcompete**
**methanogens**
Sulfate concentrations in most groundwater samples were low (3.2 mg/L to 11 mg/L)
(Supplementary Table S2 online). Groundwater samples D and H were higher with 55
mg/L and 29 mg/L, respectively (Supplementary Table S2 online). Sequence and
functional *dsrA* gene analysis (encoding the dissimilatory sulfite reductase of SRB)
revealed that SRB are present in all groundwater samples at relatively high abundances (5
- 10 % of the overall microbial community; Figures 2 and 3). These SRB are potentially
outcompeting methanogenic archaea for substrates such as acetate and $H_2$. Sulfate
concentrations higher than 3 mg/L, as detected in all groundwater samples (3.2 mg/L – 55





mg/L), could potentially create a $SO_4^{2-}$ -reducing environment with the predominance of
SRB over methanogens. This would maintain the acetate at concentrations too low for
methanogens to grow (Lovley et al., 1985). Deltaproteobacteria were dominant in all
groundwater samples, and most of the sequences were closely related to acetate-oxidising,
sulfate/sulfur-reducing    bacteria    (*Desulfovibrionales*,    *Syntrophobacterales,*
*Desulfuromonadales*; Figure 3). Additionally, *Methylocella* spp. are capable of using
methanogenic substrates, such as acetate and methylamines, for their metabolism and
therefore are not limited to growing on one-carbon compounds such as $CH_4$ (Dedysh et
al., 2005). This could have major implications for the lack of methanogenic activity in the
groundwater.

**4.4      Microbial methane oxidation in the groundwater catalyses upward migrating**
**methane from the WCM**
The functional gene for aerobic $CH_4$ oxidation (*pmoA*) was detected at relatively high
concentrations ($7.9 \times 10^2$ - $9.3 \times 10^3$ targets/mL) compared to the overall bacterial 16S
rRNA concentration ($2.5 \times 10^4$ - $5.1 \times 10^4$ targets/mL) (Figure 2). All groundwater
samples were characterised with regard to the community structure of MOB. The samples
harboured a low-diversity methanotrophic community associated with the order
*Rhizobiales* ($\alpha$-Proteobacteria), however MOB accounted for up to 7 % of the overall
microbial community (Figure 3). All groundwater samples were dominated by two MOB,
belonging to the type II methanotrophs (Figure 3). Five samples had both *Methylocella*
*palustris* (family *Beijerinckiaceae*)  and  *Methylosinus  acidophilus*  (family
*Methylocystaceae*) (samples B, D, F-H), whilst the remaining samples comprised
*Methylosinus acidophilus* only (samples A, C and E) (Figure 3). These genera were
characterised as aerobic $CH_4$ oxidisers, however aerobic MOB have been previously





observed in micro-aerophilic and anaerobic environments (Bowman, 2000). This suggests
the existence of an alternative pathway for aerobic $CH_4$ oxidation in a suboxic/anaerobic
environment. Both species have previously been found and isolated from a variety of
freshwater habitats and *Methylosinus* spp. are known to be dominant methanotrophic
populations in groundwater (Bowman, 2000). *Methylocella* and *Methylosinus* spp. possess
a soluble $CH_4$ monooxygenase (*mmoX*) (McDonald et al., 1995; Murrell et al., 2000),
which is consistent with the high abundance of the *mmoX* gene targeted in all
groundwater samples (Figure 2). Interestingly, no *pmoA* gene, a biomarker for all MOBs,
has previously been detected in known *Methylosinus* spp. (Dedysh et al., 2005). This is
supported by our data, which shows the sole predominance of *mmoX* genes in 3 of the 8
groundwater samples that are exclusively dominated by *Methylosinus* sp. (samples A, C,
and E) (Figures 2 and 3).

In addition to low concentrations of $CH_4$ reported in Iverach et al. (2015), the

dissolved $O_2$ (DO) in our groundwater samples had a large range, from low to close to
saturation (0.91 mg/L to 8.6 mg/L). *Methylocella* spp. are not associated with the
previously known type II cluster of methanotrophs, but are closely related to a non-
methanotroph (Dedysh et al., 2005) suggesting different affinities to $CH_4$ and $O_2$,
compared to previously known type II methanotrophs (Amaral and Knowles, 1995).
There is no correlation between the methanotrophic community in each sample and the
$CH_4$ data reported in Iverach et al. (2015), nor is there any correlation between the
composition of methanotrophs and DO in the groundwater (Supplementary Table S2
online).

The sample with the most diverse bacterial community (Sample F; Figure 3) had the

most [13]C-enriched individual $\delta^{13}$C-$CH_4$ relative to regional background (Iverach et al.,
2015) (Supplementary Table S3 online). A relatively high abundance (11 %) of relatives



belonging to the Chloroflexi phylum was observed exclusively in this groundwater
sample. This suggests that there are potential metabolic processes involved, such as the
microbial conversion of denitrification products to nitrogen and oxygen, that are able to
gain oxygen to facilitate the oxidation of $CH_4$ (Ettwig et al., 2010).

**4.5    Absence of AOM**
The lack of detection of the *mcrA* gene does not only indicate the absence of methanogens
but also suggests the absence of anaerobic methanotrophs (Hallam et al., 2003). Details
on the functional genomic link between methanogenic and methanotrophic archaea are
discussed comprehensively in Hallam et al. (2003). Additionally, no sequences belonging
to ANME-SRB clades were detected in the groundwater samples, indicating the absence
of ANME activity. However, members of the phylum Thaumarchaeota dominated the
archaeal community in the groundwater (Figure 3). Thaumarchaeota contains several
clusters of environmental sequences representing microorganisms with unknown energy
metabolism (Pester et al., 2011). Members of the Thaumarchaeota encode
monooxygenase-like enzymes able to utilise $CH_4$, suggestive of a role in $CH_4$ oxidation.

Samples D and H had $SO_4^{2-}$ concentrations of 55 mg/L and 29 mg/L, respectively.

This suggests that the $SO_4^{2-}$ concentration is high enough to support $SO_4^{2-}$-mediated AOM
at these sites (Whiticar, 1999). The observed $[SO_4^{2-}]$ was high enough in these 2 samples
to be able to measure the stable isotopes in the $SO_4^{2-}$. This is useful because the isotopes
yield a unique signature when $SO_4^{2-}$ reduction is coupled to $CH_4$ oxidation in anaerobic
conditions (Antler et al., 2015). However, because there are only two data points
(Supplementary Table S2 online), determining a correlation between $\delta^{34}S$-$SO_4$ and $\delta^{18}O$-
$SO_4$ is statistically invalid. The highest relative abundance of methanotrophs was found in



samples D and H (Figure 3); however, these methanotrophs are not anaerobic oxidisers
and therefore the correlation may not imply causation.

The concentration of $NO_3^-$ and $NO_2^-$ in the groundwater was also very low, with

$[NO_3^-]$ ranging from 1.2 mg/L to 2.3 mg/L and for all samples $NO_2^-$ was below 0.05 mg/L
(Supplementary Table S2 online). Therefore, AOM coupled to denitrification is unlikely
to be occurring in the groundwater of the CRAA (Nordi and Thamdrup, 2014).

The $\delta^{13}$C-DIC data indicates limited $^{13}$C-depletion as a result of DIC formation

during AOM. Segarra et al. (2015) showed that maximum $^{13}$C-depletion of DIC in the
zone of maximum AOM activity (0 – 3 cm) was highly dependent upon the isotopic
composition of the DIC before biological consumption. However, the difference between
maximum $^{13}$C-depletion of DIC and $^{13}$C-enrichment often exceeded 10‰. As our samples
are taken from deep in the aquifer (30 m or more below the ground surface), and the
difference between our most $^{13}$C-depleted DIC value and the most $^{13}$C-enriched was only
4‰ (Sample H; Supplementary Table SI online) it is unlikely that AOM is occurring in
the groundwater. Additionally, a previous study of the GAB geochemistry showed that
$\delta^{13}$C-DIC values in this region are in the range -15‰ to -6‰ (Herczeg et al., 1991). All of
our samples fall within this regional range, and we see no obvious $^{13}$C-depletion of DIC in
the groundwater that can be ascribed to AOM.

Therefore, any oxidation occurring in the groundwater would have been facilitated

by the two members of type II methanotrophs that we identified in the microbial
community analysis. Both of the species identified are classified aerobic $CH_4$ oxidisers,
agreeing with our geochemical data that no anaerobic oxidation was occurring. Despite
abundant $SO_4^{2-}$ in 2 sample locations, the absence of anaerobic methanotrophic archaea
amongst other geochemical evidence (denitrification processes) suggests that it is unlikely
that AOM is occurring within the aquifer.




**5      Conclusion**
We used geochemical and microbiological indicators to explain the occurrence of $CH_4$ in
the groundwater of an alluvial aquifer. Microbial community analysis and geochemical
data were consistent with respect to a lack of methanogenic archaea and methanogenic
activity in the aquifer. What is the original source of the $CH_4$ if not biologically produced
*in-situ*? One hypothesis to explain the presence of $CH_4$ despite there being no evidence of
methanogenesis is that there is localised upward migration of $CH_4$ from the WCM into the
CRAA via natural faults and fractures (Iverach et al., 2015).

Our geochemical data and microbiological community analysis both indicate that

AOM is not a major oxidation process occurring in the CRAA. However, the
microbiological data suggest the presence of aerobic $CH_4$ oxidisers. Due to the absence of
methanogenesis, the oxidation of $CH_4$ (facilitated by the aerobic methanotrophs present in
the groundwater) would require a secondary source of $CH_4$. The upwards migration of
$CH_4$ from the underlying WCM is the likely source.

Methane occurs naturally in groundwater, is produced via numerous biological

pathways, and can migrate through natural geological fractures. Therefore, determination
of the source of $CH_4$ using [$CH_4$] and $\delta^{13}C$-$CH_4$ data alone doesn't discern all the
processes occurring. Our microbiological community analysis showed that there were no
methanogens present to produce the $CH_4$ measured in Iverach et al. (2015) and our
geochemical analyses supported the absence of methanogenesis in the alluvial aquifer.
Similarly, the geochemical and microbiological data revealed that oxidation may not have
as large an effect on the $CH_4$ due to the low abundance of aerobic oxidisers and the
absence of anaerobic archaea.



Therefore, we suggest that the $CH_4$ detected in the CRAA in Iverach et al. (2015) is
from the local upward migration of gas from the underlying WCM, through natural faults
and fractures. A consideration of both geochemical and microbiological analyses is
particularly important in this study area because of the immediate proximity of the
underlying WCM and the proximity of the study area to CSG production. This research
uses biogeochemical constraints on the origin of $CH_4$ in a freshwater aquifer to
demonstrate the upward migration of $CH_4$ from an underlying coal seam.
**Author Contributions**
Experimental conceptualisation and design was carried out by D.I.C. & B.F.J.K.
Fieldwork was conducted by C.P.I., S.B., D.I.C. & B.F.J.K. Geochemical analyses were
conducted by D.I.C. Microbiological analyses were conducted by S.B., C.P.I. & M.M.
The manuscript was written by C.P.I. and S.B. with input from all authors.

**Acknowledgements**
This research was funded by the Cotton Research and Development Corporation and the
National Centre for Groundwater Research and Training (funded by the Australian
Research Council and the National Water Commission).
**Competing Interests**
The authors declare that they have no conflict of interest.
**List of Figures**
**Figure 1.** Site map showing the extent of the study area and sample locations within the
Condamine Catchment, south-east Queensland, Australia. Map created in QGIS; data and



imagery: Statem Toner, Open Street Map and contributors, CC-BY-SA (QGIS, 2015).
Modified with Corel Painter 2015 (Corel Corporation, 2015).
**Figure 2.** Total cell concentration and copy number abundances of bacterial and archaeal
16SrRNA genes and functional key genes for aerobic $CH_4$ oxidation (*pmoA* and *mmoX*
genes), $CH_4$ production (*mcrA* gene) and sulfate reduction (*dsrA* gene) in the groundwater
carried out by quantitative (q)PCR. Low abundances are highlighted in light blue. High
abundances are highlighted in dark blue.
**Figure 3.** Bacterial, archaeal, and methanotrophic community profiles and relative
abundances detected by Illumina sequencing.
**Figure 4.** (a) A plot of $\delta^{13}$C-DOC vs. $\delta^{13}$C-DIC, highlighting the absence of correlation
between these geochemical data, indicating that there is no methanogenic end member in
our samples. Samples E, G and H are omitted because they were below the detection limit
for $\delta^{13}$C-DOC (Supplementary Table S1.). Arrow 1 delineates the expected trend for
methanogenesis and arrow 2 is the expected trend for the dissolution of marine carbonates
(Currell et al., 2016). Arrows 3-6 highlight expected ranges for $\delta^{13}$C-DIC that are off the
scale of the graph (Currell et al., 2016). (b) A plot of $\delta^{18}$O-$H_2O$ vs. $\delta^2$H-$H_2O$ showing that
there is no $^2$H-enrichment in any of the groundwater samples. The GMWL (Craig, 1961)
and LMWL (Hughes and Crawford, 2012) are also displayed.

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
