# Peer review of "Biogeochemical constraints on the origin of methane in an alluvial aquifer: evidence for the upward migration of methane from a coal seam."

_Biogeosciences, 2016_

## Short Comment (SC1) · 19 Sep 2016

This paper uses geochemical data, microbial DNA analyses and $\delta$13C-CH4 of free gas measured in well headspaces to investigate potential production, degradation and migration processes of CH4 in the Condamine Alluvium (CRAA), which is a large agriculturally-important groundwater resource. This alluvium overlies a coal-bearing formation, the Walloon Coal Measures, which contains commercial quantities of coal seam gas (CSG) in some areas.

The paper is generally well written and the microbial DNA data is novel for this study

area. However, there are some shortfalls both with respect to information and interpretations (or lack thereof) of existing published work, as well as the interpretations of the authors' data set, some of which carry over from a preceding study (Iverach et al. 2015).

As a result, my opinion is that the major finding of the study, "that a combination of geochemical data ([CH4], [SO42-], [NO3-], [NO2-], $\delta$13C-CH4, $\delta$13C-DIC, $\delta$13C-DOC and $\delta$2H-H2O), as well as characterisation of microbiological communities present, can inform the discussion surrounding the occurrence of CH4, and its potential for upward migration in the groundwater of the CRAA", has not been achieved in the paper. In its current form, the broader interpretations of aquifer connectivity are not supported by existing data sets. This has potential to cause confusion not only in the scientific disciplines but also in the public arena (aquifer connectivity in this catchment is controversial). I suggest the discussion points below are worth considering and that some of the interpretations are revisited. I have provided additional references that may help the authors with this.

I also suggest the publication title needs revising: "....methane from a coal seam". The authors actually mean the coal measures, which contain multiple discontinuous coal seams.

1. Addressing a recent paper published in the study area.

Another paper, which I co-author (Owen et al., 2016), was very recently published (31 August) in the same journal as the co-authors' more recent paper (Iverach et al., 2015), just after Iverach et al. submitted this Biogeosciences paper (25 August). This recent paper by Owen et al. provides some discussion and alternative thinking on the interpretations made by Iverach et al. (2015). This is relevant because Iverach et al. maintain some of these interpretations from the earlier paper in this Biogeosciences paper. While the Owen et al. (2016) paper was published after the Iverach et al. submission of this Biogeosciences paper, some issues raised in the Owen et al. (2016)

paper should be addressed by Iverach et al. in this Biogeosciences paper in a revision going forward.

Owen et al. (2016) made an attempt to develop a conceptual model of CH4 in this system which could be built upon. I also encourage the authors to contribute to the conceptual understanding of this system so that an improved understanding of this system can be built upon over time as more studies are performed.

2. Thermogenic vs biogenic CH4.

Generally, the authors maintain that there is a significant thermogenic CH4 component in the coal measures. However, this is at odds with published work in this basin (Draper and Boreham, 2006; Hamilton et al., 2012; Golding et al., 2013; Hamilton et al., 2014; Baublys et al., 2015). Similarly, the recent Owen et al. (2016) paper presents high and positive $\delta$13C-DIC for CH4 in the CSG reservoir which is indicative of CO2 reduction via microbial activity. I encourage the authors to familiarise themselves with this body of work and references therein.

Two other references (Papendick et al., 2011; Hamilton et al., 2012) (lines 399 – 402, page 19) are not really supporting the authors claim of thermogenic CH4 and should be removed. In the abstract, and other parts, Hamilton et al. (2014) make clear statements about microbial CH4. Papendick et al. (2011) provides some literature review of the $\delta$13C-CH4 from the basin, and in this section Papendick et al. make a clear statement that Walloon CH4 is mainly biogenic (see section, 1.2, page 124 of Papendick et al. (2011)).

There is value in understanding CH4 origins (biogenic vs thermogenic) in CSG studies that are concerned with the gas reservoir itself because this can inform us about a number of hydrogeological characteristics. Outside the gas reservoir, however, if it is not possible to accurately distinguish thermogenic vs biogenic sources then attempts to describe thermogenic vs biogenic components are not helpful for understanding potential gas migration and it can become a distraction. This underpins the next point.

The Iverach et al. data set does not lend itself to distinguishing between thermogenic and biogenic CH4 components. While the $\delta$13C-CH4 are within a thermogenic or mixed range, it is not possible to determine thermogenic vs biogenic components solely using $\delta$13C-CH4 data. Whiticar (1999) (which Iverach et al. have referenced) provides the well-known $\delta$2H-CH4 vs $\delta$13C-CH4 plot – from the boundaries on this plot you can see that the biogenic CH4 can occur across a wide range of depleted and enriched $\delta$13C-CH4 values. However, Kotelnikova (2002), which Iverach et al. have also referenced, states in the abstract "Literature data shows that CH4 as heavy as -40‰ or -50‰ can be produced by the microbial reduction of isotopically heavy CO2"; therefore, this Whiticar plot alone is still insufficient to determine thermogenic components. Whiticar (1999) also goes into some examples of additional techniques that need to be used to identify thermogenic vs biogenic components, including the use of the Bernardi parameter, which is the ratio of C1 / C2 + C3 gases. The Whiticar paper also discusses how other processes such as oxidation and acetate fermentation can also produce $\delta$13C-CH4 that is within a similar range to thermogenic CH4. There is an excellent paper by Golding et al. (2013) that specifically reviews and discusses techniques for understanding CSG CH4 origins, including the use of isotope fractionation factors, isotopes and the Bernadi parameter. The Iverach et al data set does not lend itself to any of these analytical techniques because they have only measured $\delta$13C-CH4 and CH4 ppm in free gas samples. While the $\delta$13C-DIC in water is measured, I am not sure that a carbon isotope fractionation factor between dissolved carbon and free CH4 would make much sense here (what's needed is the $\delta$13C-CH4 of the intermediate, i.e. the $\delta$13C-CH4 of the CH4 in the water prior to any degassing).

By reconsidering the initial position on thermogenic CH4 in the study, as per the references above, then it could be argued that significant changes to data interpretations and conclusions in this paper are required. Since the authors do not have the data to assess thermogenic vs biogenic components, descriptions and interpretations on thermogenic vs biogenic CH4 should be avoided in their paper.

3. Limitations on using $\delta$13C-CH4 as "signatures".

There is a dependency in this paper on the $\delta$13C-CH4 results to interpret CH4 origins. This includes the claim that $\delta$13C-CH4 provides a "signature" of CH4 origin and source (for example, lines 105 – 108, page 5; lines 114-116 page 5; lines 199-2-1, page 9; lines 333-341, page 14-15; lines 399-401, page 19; lines 406-407, page 19). The use of the term "signature" implies a unique identifier that is distinct from all others. However, the notion of a $\delta$13C-CH4 "signature" of CH4 is not valid because there are numerous influences that can affect the isotope and associated fractionation processes. Thus, similar $\delta$13C-CH4 values can be produced coincidently, and they are not necessarily an indication of related origins/sources. The authors acknowledge this to a small degree at lines 65-67 page 3, but the issue is later ignored in their interpretations. The $\delta$13C-CH4 can behave non-conservatively under many scenarios. I suggest the authors remove all instances of the word "signature" as it is misleading, particularly to the public.

I recommend the authors familiarise themselves with Golding et al. (2013) and Owen et al (2016) and references therein, when revising their interpretations of $\delta$13C-CH4. Figure S1 in Owen et al. (2016) provides a graphical example of overlapping $\delta$13C-CH4 based on examples from published literature; these papers and their results would also be of value to these authors.

4. Identifying the $\delta$13C-CH4 coal measure end member.

Iverach et al. claim that "the $\delta$13C-CH4 of the underlying WCM in the study area has been characterised (Papendick et al., 2011; Hamilton et al., 2012; Hamilton et al., 2014)." This is a very misleading statement and it is not supported by the references provided.

Papendick et al. assessed microbial growth from samples, and do not report $\delta$13C-CH4 data. Hamilton et al. (2012) assessed gas content trends and do no report $\delta$13C-CH4 data. In any case, the Hamilton et al. (2012) study only took a handful of samples that

are possibly underlying the western alluvial edge, all of which are downstream (at least 50 km) of the Iverach et al study area.

Hamilton (2014) is the only reference used above that reports $\delta$13C-CH4 data. Based on the map provided in that study, the wells sampled for that study do not appear to be underlying the alluvium.

Commercial quantities of gas only occur in certain areas; usually geological structures where conditions support gas trapping; some useful references are (Exon, 1976; Day et al., 2006; Day et al., 2008; Day, 2009; Hamilton et al., 2012; Cook and Draper, 2013; Golding et al., 2013; Hamilton et al., 2014; Baublys et al., 2015). Generally, the gas reservoirs, where commercial quantities of gas occur, are much deeper (300-500 m+) than the alluvial-coal measures interface ($\sim$160-180m).

Importantly, the entire Walloon Coal Measures is not one big gas reservoir with homogenous $\delta$13C-CH4 and gas content. Inside gas reservoirs, where gas is trapped, conditions support enriched $\delta$13C-CH4 values. Outside the gas reservoir, conditions are different and so are the influences on $\delta$13C-CH4 fractionation. CH4 is not consistent within the coal measures, and in some places it is absent altogether (or at least below DL). Owen et al. (2016) found that many shallow coal measure wells, including some directly underlying the alluvium, contain no CH4 > DL (10 $\mu$g/L).

To my knowledge, Owen et al. (2016) presents the only published $\delta$13C-CH4 data for the coal measures that directly underlie the alluvium so far (i.e. that are outside the gas reservoirs). This shows a depleted $\delta$13C-CH4 value (-80‰ to -65‰ in the coal measures directly underlying the alluvium, which is very different to the enriched $\delta$13C-CH4 range ($\sim$ -55‰ to -47‰ that Iverach et al. are using as the coal measure end member. Owen et al. (2016) has a discussion on the changes in $\delta$13C-CH4 throughout the coal measures, and presents a conceptual model based on that data. Based on this data, I suggest that the interpretations of aquifer connectivity using $\delta$13C-CH4 in this paper need reviewing.

[Figure]

5. Microbial activity in the alluvium.

The novel microbial DNA data are interesting and provide a valuable addition to the data in this study area. The absence of methanogenic archaea is also interesting and supports Owen et al. (2016) which found very limited occurrence of CH4 in the alluvium.

Regarding sample collection, were the water samples collected for microbial DNA filtered through 0.2 micron filters during collection (Line 232, page 10)? If so, then it is possible that this filtration has resulted in sample sterilisation. Or, is the 0.2 micron filtering referring to the biomass filtering as per line 262, page 12? As it currently reads, the 2L of water collected for DNA filtering, was pre-filtered in the field using 0.2 micron filters, which would have removed microbes.

Are the DNA data for potential methanogenic archaea focussing on acetoclastic archaea only? In other words, were archaea that are responsible for CO2 reduction potentially ignored in this study? CO2 reduction is the dominant methanogenic pathway in these aquifers (Golding et al., 2013; Hamilton et al., 2014; Baublys et al., 2015; Owen et al., 2016).

It would also be beneficial to attempt to grow microbes from alluvial water samples in the laboratory: until this research is performed at sites where CH4 is measured in the water then conclusions about the general absence of methanogenic activity in this aquifer are premature.

I also wonder if the DNA data are useful for understanding aquifer connectivity. The authors use the microbial DNA data to show the absence of methanogens in the alluvium, but in Iverach et al. (2015) the authors use the same $\delta$13C-CH4 data to infer hydraulic connectivity between the aquifers at some sites. If hydraulic connectivity is occurring, wouldn't the alluvium contain the microbial consortia from the coal measures? In this Biogeosciences paper, the authors only refer to CH4 migration, which may occur independently of water migration. Are the authors revising, or suggesting a

change to, interpretations of hydraulic connectivity made in Iverach et al. (2015) in this new paper?

In order to support the claim that there is no methanogenic activity in the alluvium, Iverach et al. also use the absence of $\delta$13C-DIC enrichment and $\delta$13C-DOC depletion (Kotelnikova, 2002) to support the conclusion that there is no in situ methanogenesis in the alluvium. However, in this heterogeneous aquifer there are likely to be multiple sources of DIC and DOC. A relationship between $\delta$13C-DIC and $\delta$13C-DOC under methanogenic conditions would only be obvious in areas where high rates of methanogenesis are occurring and/or in discrete zones where single sources of DIC and DOC occur. The wells sampled by Iverach et al. are screened at multiple intervals over relatively large depth profiles. These samples can be expected to have variable (mixed) $\delta$13C-DIC based on the source/depth origin of the water and other hydrochemical conditions. Carbonates and calcretes are also common in this aquifer, and their dissolution and precipitation would affect the $\delta$13C-DIC, as would different recharge sources and any connate water. In addition, Owen et al (2016) provide a discussion on the possibility of multiple sources of DOC in the alluvium.

The authors claim "Our geochemical data also showed no evidence for the occurrence of methanogenesis in the groundwater" (Line 376-377 page 17). Arguably, this is because their geochemical data set is not capable of making this assessment conclusively. There are no CH4 measurements taken from the water itself, so how can the microbial response in the water be compared to a potential methanogenic pathway? Samples taken from wells with multiple screens over different depth profiles are likely to mask any depth-related methanogenic/redox conditions (e.g. changes in SO4 concentrations). Owen et al. (2016) showed an inverse relationship between SO4 and CH4 in the water for alluvial samples in the same aquifer where CH4 was detected. This is typical of methanogenic activity. Iverach et al. only have allegedly degassed, free CH4 data. Humez et al. (2016) provides a very good methodological approach for investigating if CH4 has potentially migrated from somewhere else or was generated in

situ.

6. $\delta$13C-CH4 are similar to background atmospheric CH4.

The CH4 ppm and $\delta$13C-CH4 measured by Iverach et al. (2015) which are used in this study are similar to background, atmospheric CH4 ppm and $\delta$13C-CH4 (Dlugokencky et al., 2011; Stalker, 2013). Iverach et al. (2015) reference these references. Owen et al. (2016) raise the issue that Iverach et al. (2015) have possibly only measured atmospheric CH4 that has been sucked into the well head during pumping. Iverach et al. are referred to that discussion in that paper.

What is missing in the Iverach et al. $\delta$13C-CH4 data set is the crucial CH4 in the water (end member), i.e. what is the $\delta$13C-CH4 prior to any degassing? What is the concentration of CH4 in the water? What is the flux of any degassing CH4 from the water and how does this affect concentration and isotope fractionation, and how does this compare between samples? Without this additional data it is not possible to confidently determine where the measured free CH4 presented by Iverach et al. (2015), and which is used again by Iverach et al. in this paper, has come from. The fact that the values are similar to background atmospheric free gas provides a plausible argument against both the findings of Iverach et al (2015) and Iverach et al. presented in this paper. Iverach et al. (2015) and Iverach et al. in this paper claim that the CH4 must be coming from the underlying coal measures; however, this is an assumption that is not yet supported by any data. The authors should be careful not to open themselves up to criticism of subjective bias, i.e. a preconception that aquifer connectivity/gas migration must be occurring. Owen et al. (2016) proposes a minimum set of parameters required to make assessments of CH4 migration. This includes discussion on the value of $\delta$2H-CH4 in assisting with determining CH4 origins.

7. Inferred vs. measured $\delta$13C-CH4 values.

At times throughout the paper, Iverach et al. report $\delta$13C-CH4 values, not as the measured values but as the values taken from a regression line as reported in Iverach
et al. (2015); for example, lines 406-407, page 19. This is very misleading for the reader.

The keeling-style plot used in Iverach et al (2015) is inappropriate because; 1) there are three potential sources of CH4 (alluvial, coal measure and atmospheric) and a keeling-style plot cannot be used to differentiate the mixing components between these three sources; 2) one of the regression lines relies on a single outlying point (removing this outlier would invalidate the regression line); 3) most of the data are clumped around similar values (is the variability in the data set significant/relevant?); 4) the method used (a combination of DOC and 3H) to categorize samples is questioned. Iverach et al. are referred to this discussion in Owen et al. (2016) for more detail.

The discussion by Iverach et al. on CH4 origins tends to be assumptive and at times not logical. For example, at one point the authors claim that three samples "have a typically biogenic isotopic source signature (-69.1‰" (lines 406-407 page 19): recall that this is not the measured $\delta$13C-CH4, but the $\delta$13C-CH4 inferred from the regression line in Iverach et al. (2015); the measured values are $\sim$ -47‰. The possible biogenic sources are explained as being coal measure-derived CH4 from an area where biogenic CH4 has replaced thermogenic CH4 in the coal measures. Biogenic CH4 can replace thermogenic CH4, but then this discussion point is inconsistent with the authors' earlier thermogenic descriptions of the basin/coal measure CH4. If thermogenic and biogenic CH4 components change throughout the coal measures, then it would be reasonable to expect the proportion of biogenic vs thermogenic CH4 in any migrating gas to also vary, and mixing would then also occur in the alluvium. Acknowledging this adds a great deal of complexity to this type of study which is not addressed in this paper.

In this paper, it would be more transparent for Iverach et al. to report the actual measured $\delta$13C-CH4 values at all times. Given that Owen et al. (2016) raise discussion points about the use of the Iverach et al. (2015) methods, this Biogeosciences paper should either address those points, or avoid relying on the Iverach et al. (2015)

$\delta$13C-CH4 results altogether. Given the isotopic data of the shallow coal measures underlying the alluvium (a biogenic, depleted $\delta$13C-CH4 range) presented in Owen et al. (2016), the interpretations made by Iverach et al. (2015) and by Iverach et al. in this paper could be questioned (see discussion point 4 above).

8. No samples were taken from the allegedly discharging aquifer.

A major shortfall of this paper and of Iverach et al. (2015) is that not a single sample was taken from the allegedly discharging aquifer, the Walloon Coal Measures. Samples have only been taken from the alluvium. It is without precedent that conclusions regarding hydraulic connectivity and aquifer interactions have been drawn in two successive hydrogeological studies without using appropriate reference samples from both the allegedly interacting aquifers.

In order to make valuable use of their data within the appropriate scientific limitations, I would encourage Iverach et al. in this paper to avoid jumping to conclusions about aquifer connectivity. I would recommend the authors assess the data that they have, play to its strengths with respect to the data limitations, and focus on contributing to the understanding of this system rather than resolving broader questions about aquifer connectivity, which cannot be confidently made from this data set.

9. Incorrect referencing.

In addition to the incorrect references regarding $\delta$13C-CH4 and thermogenic vs biogenic CH4 identified in the above discussion points, there are other cases where Iverach et al. have incorrectly referenced material or have misrepresented quotes or statements from referenced material. This lets the paper down. I have provided some examples below, with discussion points and recommendations for new references in places. I encourage the authors to revise their work and check references carefully and present transparent, objective discussion based on all available literature.

Geological/hydrogeological references: In a number of places incorrect references are

used when discussing background information. For example Hillier (2010) (Line 170-172, page 8) and KCB (2011) (Line 174-176, page 8) are non-peer-reviewed consultant reports that did not present the background information being referred to. Similarly, Kelly and Merrick (2007) (Line 178-181, page 8), is also not peer-reviewed and summarises other published work. Authors are referred to appropriate literature that provides the background information for referencing (e.g. Lane, 1979; Huxley, 1982; Cook and Draper, 2013).

Inappropriate references: Kelly et al. (2014) (Line 178-181, page 8) and Duvert et al. (2015) (Line 197-199, page 9) pertain to studies in two different catchments (the Namoi and Teviot Brook catchments, respectively). These references are inappropriate to reference here.

Misleading referencing: "Huxley (1982) and Hillier (2010) both suggest that the general decline in water quality downstream is due to some net flow of the more saline WCM water into the CRAA." (Line 193-195, page 9).

The theory that a general decrease in water quality downstream is due to an influx of more saline water from the coal measures has been examined in Owen and Cox (2015). That paper found that evapotranspiration was the dominant influence on salinity in the alluvium, and that the most saline water occurs in the shallowest alluvial wells. An increase in clay content and decrease in hydraulic conductivity downstream of Cecil Plains, which are characteristics identified by Huxley (1982), would be helping to drive these processes. Since Iverach et al. (2015) and Iverach et al. in this paper both reference Owen and Cox (2015) already, it is not clear why that paper is ignored in this context. Non-peer reviewed material (e.g. Hillier, 2010) should not be used in place of more recent peer-reviewed material.

"Duvert et al. (2015) and Owen and Cox (2015) both used hydrogeochemical analyses to show that there was limited movement of water between the two formations." (Line 197-199, page 9)

Owen and Cox (2015) did not explicitly assess water movement between the two aquifers: the statement above is misleading. It is frustrating to see this material referenced in this way, particularly as Iverach et al. (2015) already also misquote this paper. The aims of Owen and Cox (2015) were to investigate hydrochemical evolution within the alluvium, with a focus on the origins of Na and Cl and Na-HCO3 water types. This is clearly outlined in the introduction of that paper. The conclusions were that the dominant water types in the alluvium were the result of other processes not related to bedrock discharge, however the paper also noted in the conclusions that "The persistence of Na–HCO3–Cl water types in some deep alluvial wells at the alluvial–WCM interface from Cecil Plains to Dalby provides the most likely indication of any alluvial–WCM interaction; however, these water types also occur in shallower alluvial zones." Clearly this paper does not make definitive conclusions regarding aquifer connectivity, and it even highlights areas where potential aquifer interaction could be occurring. It appears that Iverach et al. have not read Owen and Cox (2015) properly. It is not appropriate to continue to misrepresent this published work, and to continue to rely on earlier, non-peer reviewed work (e.g. Hillier, 2010) to support particular angles/arguments.

References

BAUBLYS, K. A., HAMILTON, S. K., GOLDING, S. D., VINK, S. & ESTERLE, J. 2015. Microbial controls on the origin and evolution of coal seam gases and production waters of the Walloon Subgroup; Surat Basin, Australia. International Journal of Coal Geology, 147–148, 85-104.

COOK, A. G. & DRAPER, J. J. (eds.) 2013. Geology of Queensland, Brisbane, QLD: Geological Survey of Queensland.

DAY, R. 2009. Coal seam gas booms in eastern Australia. Preview: Australian Society of Exploration Geophysicists, 140, 26-32.

DAY, R. W., BUBENDORFER, P. J. & PINDER, B. J. Petroleum potential of the easternmost Surat Basin in Queensland. Proceedings of the petroleum Exploration Society of

Australia Eastern Australasian Basins Symposium III, 2008 Sydney. Melbourne, Victoria: Petroleum Exploration Society of Australia 191-199.

DAY, R. W., PREFONTAINE, R. F., BUBENDORFER, P. A. J., OBERHARDT, M. H., PINDER, B. J., HOLDEN, D. J. & GUNNESS, R. A. 2006. Discovery and development of the Kogan North and Tipton West Coal Seam Gas (CSG) Fields, Surat Basin, Southeast Queensland. APPEA Journal, 46, 367-381.

DLUGOKENCKY, E. J., NISBET, E. G., FISHER, R. & LOWRY, D. 2011. Global atmospheric methane: budget, changes and dangers. Philosophical Transactions of the Royal Society of London A: Mathematical, Physical and Engineering Sciences, 369, 2058-2072.

DRAPER, J. J. & BOREHAM, C. J. 2006. Geological controls on exploitable coal seam gas distribution in Queensland. APPEA Journal, 46, 343–366.

EXON, N. F. 1976. The Geology of the Surat Basin: Bulletin 166. In: RESOURCES, D. O. N. (ed.) Bureau of Mineral Resources, Geology and Geophysics. Canberra, ACT: Australian Government Publishing Service.

GOLDING, S. D., BOREHAM, C. J. & ESTERLE, J. S. 2013. Stable isotope geochemistry of coal bed and shale gas and related production waters: A review. International Journal of Coal Geology, 120, 24-40.

HAMILTON, S. K., ESTERLE, J. S. & GOLDING, S. D. 2012. Geological interpretation of gas content trends, Walloon Subgroup, eastern Surat Basin, Queensland, Australia. International Journal of Coal Geology, 101, 21-35.

HAMILTON, S. K., GOLDING, S. D., BAUBLYS, K. A. & ESTERLE, J. S. 2014. Stable isotopic and molecular composition of desorbed coal seam gases from the Walloon Subgroup, eastern Surat Basin, Australia. International Journal of Coal Geology, 122, 21-36.

HUMEZ, P., MAYER, B., NIGHTINGALE, M., BECKER, V., KINGSTON, A., TAYLOR,

S., BAYEGNAK, G., MILLOT, R., KLOPPMANN, W. 2016. Redox controls on methane formation, migration and fate in shallow aquifers. Hydrol. Earth Syst. Sci. Discuss.

HUXLEY, W. J. 1982. The hydrogeology, hydrology and hydrochemistry of the Condamine River Valley Alluvium. Masters, Queensland Institute of Technology.

IVERACH, C. P., CENDÓN, D. I., HANKIN, S. I., LOWRY, D., FISHER, R. E., FRANCE, J. L., NISBET, E. G., BAKER, A. & KELLY, B. F. J. 2015. Assessing Connectivity Between an Overlying Aquifer and a Coal Seam Gas Resource Using Methane Isotopes, Dissolved Organic Carbon and Tritium. Scientific Reports, 5, 15996.

KOTELNIKOVA, S. 2002. Microbial production and oxidation of methane in deep subsurface. Earth-Science Reviews, 58, 367 - 395.

LANE, W. B. 1979. Progress Report on Condamine Underground Investigation to December 1978. Brisbane, Queensland.

OWEN, D. D. R., SHOUAKAR-STASH, O., MORGENSTERN, U. & ARAVENA, R. 2016. Thermodynamic and hydrochemical controls on CH4 in a coal seam gas and overlying alluvial aquifer: new insights into CH4 origins. Scientific Reports, 6, 32407.

PAPENDICK, S. L., DOWNS, K. R., VO, K. D., HAMILTON, S. K., DAWSON, G. K. W., GOLDING, S. D. & GILCREASE, P. C. 2011. Biogenic methane potential for Surat Basin, Queensland coal seams. International Journal of Coal Geology, 88, 123-134.

STALKER, L. 2013. Methane origins and behaviour. Australia: Commonwealth Scientific and Industrial Research Organisation.

WHITICAR, M. J. 1999. Carbon and hydrogen isotope systematics of bacterial formation and oxidation of methane. Chemical Geology, 161, 291-314.

---

## Referee Comment (RC1) · M. Currell (Referee) · 20 Sep 2016

Iverach et al. present a novel approach to the determination of methane sources in shallow groundwater in the Condamine Alluvium aquifer, Australia. I think the study is of high scientific significance, for two main reasons:

1. The use of combined geochemical and microbiological indicators to study the origins of methane in groundwater is novel. Studies of this kind are relatively rare in the literature, and the microbiological analysis provide insight about the methane sources and degradation processes that could't otherwise be gained from the isotopic analyses

alone 2. The topic and research question(s) are of high importance, given the current debate about environmental impacts of coal seam gas (and other unconventional gas), both in this particular area of Australia, and worldwide. There are some minor issues and corrections needed, and some areas where additional information could be included to make the paper more solid. However, overall I think this is a high quality manuscript.

Specific comments: Abstract Line 33-34: Which data? I like to see some actual data values or description of the particular aspects of the data set of greatest significance (and supporting the conclusions described) included in the abstract. If more space is needed in order to do this, I suggest removing the second sentence of the abstract, as this is background information that can be included in the introduction.

Introduction Line 50: I suggest adding the term 'in situ' when discussing biological production of methane in the shallow groundwater. This makes it clear that you are distinguishing two different potential gas sources- one produced in the shallow aquifer itself, and another whereby gas from another unit has migrated to the aquifer. Line 81: 'Therefore' is not really the best word here. It does not follow logically from the preceding discussion that combining geochemistry/microbiology can discriminate the relevant processes; rather you could say that microbiological indicators have the potential to resolve some of the uncertainties just mentioned (e.g. methanogenesis and methane degradation processes), that can't be otherwise determined on the basis of geochemical data alone. Here you could also note the general absence of published studies which have combined geochemical and microbiological indicators to look at methane sources and degradation in an applied setting (an important point to make in your introduction). Line 103: See previous comment; this could be clarified by adding 'in situ methanogenesis' to distinguish from gas migration from another unit. Line 104-108: I think you should expand this paragraph and include some of the actual data, e.g. the observed ranges and mean/median values of d13CCH4 and d13CDIC found in the WCM from other published studies. This can be included in the text (e.g. ranges,

mean values etc), as well as in a table. This would help to strengthen your isotopic lines of evidence to support the hypothesised migration mechanism later in the manuscript. Note that Baublys et al 2015 (Int. J. Coal Geol v.147-8, pp85-104) have also reported extensive data on isotopic composition of gases and water in the WCM, which should be included along with other recent published studies.

Study area Line 146-47: Try to avoid repetition (primary/primarily) Line 151: Suggest adding 'including methane concentrations' at the end of this sentence, to highlight the significance of what you are looking at (mostly the methane in groundwater). 2.1 Hydrogeological setting. Could you include a cross section or at least a stratigraphic column to go with your description of the geological units? Line 160: 'The CRAA sits within the Surat Basin, which is a major sub-province of the Great Artesian Basin'. Perhaps refer to one of the Geoscience Australia and/or CSIRO hydrogeology reports on the GAB (e.g. Ransley and Smerdon, 2012). Line 188: The recent studies by the Office of Groundwater Impact Assessment (OGIA) may have more detail about the connectivity between the CRAA and the WCM and the extent of the aquitard(s), e.g. the Surat Underground water impact report (OGIA, 2016). Line 203: Connectivity for gas? water? both?

Method Line 212: Here you should refer to a figure and/or table which includes your sample depths and locations Line 233-34: Were the physico-chemical parameters (EC, pH, DO) monitored during the second round of sampling? If so, you could report these and use as evidence that the water composition between the two sampling events did not change substantially (if this is true). Line 238-239: What about cations? Line 242-243: Can you refer to a published paper where the same method was used? Same for the DIC isotopes (line 245).

Results & Discussion Line 371: Suggest writing 'in situ within the CRAA' instead of 'locally' to be clearer. Line 377: Do you mean the major ion data? Which particular aspects (e.g. sulfate and nitrate concentration data)? Line 396: Suggest changing to: 'major processes resulting in CH4 in the CRAA' rather than 'producing CH4 in the

CRAA' (or you could say 'responsible for the presence of CH4'). Line 398: Suggest changing 'coming from' to 'derived from'. Line 406-411: This paragraph is a bit confusing and needs re-writing. Is the gas in the WCM really 'typically thermogenic'? All of the isotopic data for 13CCH4 I have seen for gases and water in the WCM indicates a bacterial source of methane (e.g. 13CCH4 values around -50permil) rather than thermogenic (which should have values higher than -40permil). Is there anything else distinctive about the samples with more depleted 13CCH4, such as a much lower CH4 concentrations or differences in the major ions that could explain the isotopic difference? Line 431-432: Yes, and further, the evidence about the presence of sulfate and conditions favouring SRB is a further line of evidence that in situ methanogenesis is unlikely to be responsible for the CH4 in the shallow aquifer Line 434 - 476: The section on methane oxidation is insightful; good use of the microbiological methods to combine with the isotopic data and yield some new insights. Line 478: Use the full name for AOM in the title. Line 499-500: Relative to what? Other water in the CRAA?

General comment I think including a figure showing your isotopic compositions (13CCH4) and concentrations of methane, (using the data from Iverach 2015) and comparing with other published data on isotopic characteristics of WCM gases would be helpful, to strengthen the evidence for the proposed hypothesis (together with the microbiological indicators).

Conclusions Line 536: You could also note your other lines of evidence here (e.g., that this is supported by the co-existence of CH4 with sulfate in the groundwater, and the isotopic composition of the methane). Line 547-548: Your study does not really provide information about the precise pathway(s) by which methane migrates from the WCM to the CRAA, only strong evidence that such migration occurs. Hence, the statement about 'through natural faults and fractures' is really just speculation. Unless you can support it with some geological evidence, other mechanisms may also be responsible (such as transport along wells that are not fully sealed, direct leakage of gas between the units where the aquitard is absent). I suggest either talking about all possible path-
ways (including these), or simply leaving out the discussion of the pathway altogether and sticking to what your data shows.

---

## Referee Comment (RC2) · Anonymous Referee #2 · 17 Oct 2016

General comments

Generally, the manuscript address scientific questions within the scope of BG; proving the source of methane in shallow aquifer is a relevant and important issue. The author's present data which indicate that methane detected in an alluvial aquifer is not produced in the aquifer itself but is produced in the underlying coal seam and subsequently migrates upwards to the aquifer. This finding would be of fundamental interest for the risk assessment regarding the occurrence of methane in shallow aquifers. However, three of the authors (including the first and last author) published already in 2015 a paper in which basically the same conclusion has been drawn (Iverach et al., 2015); moreover, essential data – the carbon isotope signatures of methane – shown in the present manuscript have been already published by Iverach et al. (2015). This reduces the originality and novelty of this paper.

The overall presentation is well structured and clear, including an accurate title, a proper abstract and introduction into the topic, and adequate citations of related work.

The applied methods and assumptions are valid; some of the used scientific methods are not clearly described and cannot be reproduced (see specific comments). Generally, the results are sufficient to support the main conclusion that the source of the methane detected in the alluvial aquifer was the underlying coal seam. Some interpretations based on the geochemical and microbiological data are certainly speculative (see specific comments) and need to be supported by literature/experimental data; if not possible, these parts should be condensed or deleted. On the other hand, one important result of this study, the oxygen concentrations of the investigated groundwater samples, is not seriously presented and discussed in the main manuscript (the data are somewhat hidden in the supplemental information). The oxygen data indicate that the studied aquifer zones are predominantly aerobic, a fact that could explain the absence of strictly anaerobic methanogens in the groundwater samples. Due to the presence of methanotrophs and availability of oxygen in the aquifer, the question arises to which extent methane is oxidized and whether aerobic oxidation of methane is trackable in the aquifer by compound specific stable isotope analysis, as this reaction is characterized by strong carbon and hydrogen isotope fractionation (Feisthauer et al., 2011). Unfortunately, this aspect is not discussed in the manuscript.

Specific comments

Lines 96-103: This statement is too strict. It's true that sulfate reducers generally outcompete methanogens but not always, see Struchtemeyer et al. (2005).

Lines 119-133: I suggest mentioning that the expression of the particulate and soluble methane monooxygenase is triggered by the amount of available copper ions.

Lines 208-212: For clarity, I suggest indicating the depth at which each well was sampled. I do not understand why the eight samples are representative of the aquifer, please explain in detail.

Line 226: How long were the DIC samples stored before measurement? Please indicate.

Lines 228-230: I wonder why samples for geochemical and microbiological analyses were not sampled at the same time, which would have strengthened the main conclusions of this paper.

Lines 232: Probably, any nanobacteria (prokaryotes smaller than 0.2 μm) were lost during this procedure?

Lines 241-259: Give references for the methods of $\delta^2$H-H$_2$O, $\delta^{18}$O-H$_2$O, $\delta^{13}$C-DIC, $\delta^{13}$C-DOC, $\delta^{18}$O-SO$_4^{2-}$ and $\delta^{34}$S-SO$_4^2$ analysis or describe the methods in detail that they can be reproduced.

Lines 262 ff. A critical question is whether the microbial community of a groundwater sample will truly reflect the microbial community of the subsurface from which the groundwater was extracted from. This aspect should be briefly discussed (probably in the Results & Discussion section).

Figure 2: In the Figure, five ranges are shown (indicated by 5 different colors) whereas only four ranges are given in the legend. I recommend using different colors for each order of magnitude for higher resolution. A general drawback of Figure 2 is the lack of any statistics, what are the standard deviations of the data?

Line 420 ff. See comment above. It's true that sulfate reducers generally outcompete methanogens but not always, see Struchtemeyer et al. (2005). I recommend discussing with more caution.

Lines 425-428: It is very speculative to conclude that the detected phylotypes affiliated to sulfate or sulfur reducers will oxidize acetate (or outcompete methanogens). I suggest discussing with more caution. Deducing specific metabolic activities from partial 16S rDNA sequences is questionable.

Lines 428-432: I do not understand this argumentation. Methylocella are aerobic organisms, whether methanogens are strictly anaerobic. They probably do not exist in the same ecological niche.

Lines 448-450: What could be an alternative pathway for aerobic methane oxidation in an anaerobic environment? The initial methane oxidation reactions will always depend on molecular oxygen, hence aerobic methane oxidation cannot take place in the absence of oxygen. Why not discussing the detected (high) oxygen concentrations of the groundwater samples in this context?

Lines 460-462: I wonder why the oxygen data are not shown in more detail. Some wells seem to be fully aerobic, a result which does not correspond to the observation of the dominance of sulfate or sulfur reducing deltaproteobacteria in most of the samples. On the other hand, the presence of oxygen explains well the presence of methanotrophs and other aerobes in the groundwater samples. Probably, the discrepancy might be explained by the sampling artifacts; the pumped groundwater may contain strictly anaerobic organisms originally attached to the aquifer solids in which anoxic microenvironments exist.

Lines 470-476: This hypothesis is very, very speculative. Are there any indications for the presence of nitrate in the groundwater? Why Chloroflexi should convert denitrification products to oxygen? The hypothesis needs more arguments (support by literature or own experimental data); if no other arguments are available, I suggest deleting this passage.

Lines 487-488: Give references for this statement.

Lines 490-491: I doubt that the methane concentrations were high enough to allow sulfate-dependent AOM. Please discuss.

Cited literature:

Feisthauer S, Vogt C, Modrzynski J, Szlenkier M, Krüger M, Siegert M, Richnow HH (2011) Different types of methane monooxygenases produce similar carbon and hydrogen isotope fractionation patterns during methane oxidation. Geochim. Cosmochim. Acta 75: 1173-1184

Iverach CP, Cendón DI, Hankin SI, Lowry D, Fisher RE, France JL, Baker A, Kelly BFJ (2015) Assessing connectivity between an overlying aquifer and a coals seam gas resource using methane isotopes, dissolved organic carbon and tritium. Sci. Rep. 5: 1-11

Struchtemeyer CG, Elshahed MS, Duncan KE,  McInerney MJ (2005) Evidence for aceticlastic methanogenesis in the presence of sulfate in a gas condensate-contaminated aquifer. Appl. Environ. Microbiol. 71: 5348-5353

Technical comments

Line 322: DSMZ, Braunschweig, Germany

---

## Author Comment (AC1) · 25 Nov 2016

We thank the reviewer for their time and constructive comments on our manuscript. We have addressed all concerns raised below.

**General comments**

Generally, the manuscript address scientific questions within the scope of BG; proving the source of methane in shallow aquifer is a relevant and important issue. The author's present data which indicate that methane detected in an alluvial aquifer is not produced in the aquifer itself but is produced in the underlying coal seam and subsequently migrates upwards to the aquifer. This finding would be of fundamental interest for the risk assessment regarding the occurrence of methane in shallow aquifers. However, three of the authors (including the first and last author) published already in 2015 a paper in which basically the same conclusion has been drawn (Iverach et al., 2015); moreover, essential data – the carbon isotope signatures of methane – shown in the present manuscript have been already published by Iverach et al. (2015). This reduces the originality and novelty of this paper.

The microbiological data presented in this paper are unique and vastly improve our understanding of this aquifer system. A small portion of the geochemical data from the previous manuscript was reproduced here for ease of reading the paper.

The overall presentation is well structured and clear, including an accurate title, a proper abstract and introduction into the topic, and adequate citations of related work.

The applied methods and assumptions are valid; some of the used scientific methods are not clearly described and cannot be reproduced (see specific comments). Generally, the results are sufficient to support the main conclusion that the source of the methane detected in the alluvial aquifer was the underlying coal seam. Some interpretations based on the geochemical and microbiological data are certainly speculative (see specific comments) and need to be supported by literature/experimental data; if not possible, these parts should be condensed or deleted.

We have added citations to all mentioned methods, and we have addressed the specific speculative comments below.

On the other hand, one important result of this study, the oxygen concentrations of the investigated groundwater samples, is not seriously presented and discussed in the main manuscript (the data are somewhat hidden in the supplemental information). The oxygen data indicate that the studied aquifer zones are predominantly aerobic, a fact that could explain the absence of strictly anaerobic methanogens in the groundwater samples. Due to the presence of methanotrophs and availability of oxygen in the aquifer, the question arises to which extent methane is oxidized and whether aerobic oxidation of methane is trackable in the aquifer by compound specific stable isotope analysis, as this reaction is characterized by strong carbon and hydrogen isotope fractionation (Feisthauer et al., 2011). Unfortunately, this aspect is not discussed in the manuscript.

The dissolved oxygen data in the groundwater were measured using a YSI probe on the surface that was also measuring the pH, EC, TDS, temp. As such, it is not a completely accurate representation of the DO conditions in the aquifer, as the degassing caused by pumping and the effect of the barometric pressure needs to be considered. However, we have mentioned the high DO concentration (line 541), addressing the comments above, as well as

the DO concerns raised below. Unfortunately, tracking methane oxidation was outside the scope of this study, which aimed at characterising for the first time the microbial community in this freshwater aquifer and seeing if it was possible to use microbes to help elucidate the source of $CH_4$ detected in the aquifer. It would be a very useful future study, but we have not mentioned it in the text because it is outside the scope of this investigation.

**Specific comments**

Lines 96-103: This statement is too strict. It's true that sulfate reducers generally outcompete methanogens but not always, see Struchtemeyer et al. (2005).

This statement has been softened: "…because SRB often outcompete methanogenic archaea…" and the suggested reference has been included.

Lines 119-133: I suggest mentioning that the expression of the particulate and soluble methane monooxygenase is triggered by the amount of available copper ions.

This has been mentioned at the suggested location in the text.

Lines 208-212: For clarity, I suggest indicating the depth at which each well was sampled. I do not understand why the eight samples are representative of the aquifer, please explain in detail.

A table indicating the slotted interval for each sample has been included in the methods now. We understand that eight samples are a small dataset, however they are at varying depths and locations throughout the aquifer. Physico-chemical parameters and the spread of geochemical data indicate that the samples are representative of the spread of the conditions of the aquifer as a whole.

Line 226: How long were the DIC samples stored before measurement? Please indicate.

The DIC samples were analysed within one month and this information has now been included in the manuscript. They were also filtered through a 0.22 μm filter in the field, which is the best way to maintain the sample (provided refrigeration and proper storage) (Doctor et al. 2008). In addition, DIC samples from another field site were analysed 1 week after collection, and then re-analysed 6 months later and were found to have no difference in measurement.

Lines 228-230: I wonder why samples for geochemical and microbiological analyses were not sampled at the same time, which would have strengthened the main conclusions of this paper.

Insights from the original hydrogeochemical survey indicated that microbiological data would refine our understanding of the processes.  Therefore we returned and collected microbiological data (at a limited number of sites due to budget constraints). In December of the same year (when the aquifer is under the same stress as in January), additional funding was granted and we were able to sample for the microbiology.

Lines 232: Probably, any nanobacteria (prokaryotes smaller than 0.2 μm) were lost during this procedure?

A 0.2 μm filter is standard for filtering microbial communities. The filtrate was also screened using SYBRGREEN I staining and microscopy and there was no detection of cells.

Lines 241-259: Give references for the methods of d2H-H2O, d18O-H2O, d13C-DIC, d13C-DOC, d18O-SO4, d34S-SO4 analysis or describe the methods in detail that they can be reproduced.

References for the methods of analysis for all geochemical data have been provided in this section.

Lines 262 ff. A critical question is whether the microbial community of a groundwater sample will truly reflect the microbial community of the subsurface from which the groundwater was extracted from. This aspect should be briefly discussed (probably in the Results & Discussion section).

We do believe that the microbial community of the groundwater is reflecting the microbial community of the subsurface. Maamar et al. (2015) found that the microbial community composition of groundwater was controlled by groundwater residence times and the location of samples along the groundwater flow path, independent of the geology, stating that "hydrogeologic circulation exercises a major control on microbial communities". They also state: "…Thus, geochemical conditions, and in particular the availability of electron donors and acceptors, are a major driver of microbial community composition and diversity in groundwater and the geological substratum".

Additionally, when we sample the groundwater, we are also sampling fine particles with biomass attached. Further, the Condamine production wells are drawing water that is representative of the sampled formations and the intense purging ensures that this is the case. The $^{14}$C and $^{3}$H activities suggest that we are not drawing a modern/old mixed groundwater component, therefore whatever water is sampled is representative of the formation, and we presume the microbial communities within it.

A small paragraph explaining the above has been included in the discussion (line 507).

Figure 2: In the Figure, five ranges are shown (indicated by 5 different colors) whereas only four ranges are given in the legend. I recommend using different colors for each order of magnitude for higher resolution. A general drawback of Figure 2 is the lack of any statistics, what are the standard deviations of the data?

Figure 2 has been changed - 4 different colours have been used for 4 different ranges. Standard deviations have been added to the figure legend and qPCR specific validations are in the methods (line 366-370).

Line 420 ff. See comment above. It's true that sulfate reducers generally outcompete methanogens but not always, see Struchtemeyer et al. (2005). I recommend discussing with more caution.

We have clarified the language above, however in the text at this location we do already say "These SRB are potentially outcompeting methanogenic archaea…", implying that this may not be the case. We then proceed with additional evidence as to why the lack of methanogenic archaea could be a result of this competition.

Lines 425-428: It is very speculative to conclude that the detected phylotypes affiliated to sulfate or sulfur reducers will oxidize acetate (or outcompete methanogens). I suggest discussing with more caution. Deducing specific metabolic activities from partial 16S rDNA sequences is questionable.

We have clarified our discussion. Because most of the Deltaproteobacteria sequences detected in the groundwater were closely related to acetate-oxidising sulfate/sulfur reducing bacteria (*Desulfovibrionales, Syntrophobacterales, Desulfuromonadales*), it is reasonable to assume that the lack of methanogenic archaea could potentially be a result of competition from sulfate reducers taking the acetate, which is the methanogenic substrate required.

Lines 428-432: I do not understand this argumentation. Methylocella are aerobic organisms, whether methanogens are strictly anaerobic. They probably do not exist in the same ecological niche.

Aerobic and anaerobic microorganisms can exist in the same environment. They are not strictly separated; e.g. anaerobic methanogens can occur in anoxic or suboxic microniches in mainly aerobic environments (Kato et al., 2007; Dimikić et al., 2011).

Lines 448-450: What could be an alternative pathway for aerobic methane oxidation in an anaerobic environment? The initial methane oxidation reactions will always depend on molecular oxygen, hence aerobic methane oxidation cannot take place in the absence of oxygen. Why not discussing the detected (high) oxygen concentrations of the groundwater samples in this context?

As previously mentioned, the detected high concentrations of dissolved oxygen in the groundwater have been discussed now and it has been stated that these are most likely the reason for abundant aerobic methanotrophs in the groundwater. Therefore, an alternative pathway for aerobic methanotrophs, potentially using other electron acceptors, has not been discussed.

Lines 460-462: I wonder why the oxygen data are not shown in more detail. Some wells seem to be fully aerobic, a result which does not correspond to the observation of the dominance of sulfate or sulfur reducing deltaproteobacteria in most of the samples. On the other hand, the presence of oxygen explains well the presence of methanotrophs and other aerobes in the groundwater samples. Probably, the discrepancy might be explained by the sampling artifacts; the pumped groundwater may contain strictly anaerobic organisms originally attached to the aquifer solids in which anoxic microenvironments exist.

As mentioned previously, the DO data are not a completely accurate representation of DO concentration within the aquifer - this is why they were included in the supplementary material but not highlighted in the text. If the discrepancy between DO and deltaproteobacteria is to be explained by sampling artifacts, it would probably be this, not microbial sampling methods.

Aerobic and anaerobic microorganisms can live alongside each other in many habitats in microniches. Sulfate reduction under oxic conditions has been observed and previously published; e.g. in cyanobacterial mats or periodically in activated sludge (Kjeldsen et al. 2004; Fike et al. 2008).

We have now explicitly referred to the role that the high concentration of DO is potentially playing in the absence of methanogenic archaea and abundance of aerobic bacteria (line 541). In addition, we have explained why the deltaproteobacteria are dominant in most samples despite the presence of O2.

Lines 470-476: This hypothesis is very, very speculative. Are there any indications for the presence of nitrate in the groundwater? Why Chloroflexi should convert denitrification products to oxygen? The hypothesis needs more arguments (support by literature or own experimental data); if no other arguments are available, I suggest deleting this passage.

We have removed this hypothesis.

Lines 487-488: Give references for this statement.

A reference has been given for this statement (Pester et al. 2011).

Lines 490-491: I doubt that the methane concentrations were high enough to allow sulfate-dependent AOM. Please discuss.

We agree that methane concentrations were most likely not high enough to allow sulfate-dependent AOM in this groundwater. However, at this location in the manuscript we are going step-wise through our data providing evidence either for or against potential processes affecting the occurrence of CH4 in this groundwater – at this particular point, it is the possible occurrence of AOM in the groundwater. Hence, we state that the sulfate concentrations are potentially high enough to mediate AOM at 2 locations, however, we go on to state that further geochemical evidence (including lack of detected ANME's) indicate that this process is not occurring.

Cited literature:

Feisthauer S, Vogt C, Modrzynski J, Szlenkier M, Krüger M, Siegert M, Richnow HH (2011) Different types of methane monooxygenases produce similar carbon and hydrogen isotope fractionation patterns during methane oxidation. Geochim. Cosmochim. Acta 75: 1173-1184

Iverach CP, Cendón DI, Hankin SI, Lowry D, Fisher RE, France JL, Baker A, Kelly BFJ (2015) Assessing connectivity between an overlying aquifer and a coals seam gas resource using methane isotopes, dissolved organic carbon and tritium. Sci. Rep. 5: 1-11

Struchtemeyer CG, Elshahed MS, Duncan KE, McInerney MJ (2005) Evidence for aceticlastic methanogenesis in the presence of sulfate in a gas condensate-contaminated aquifer. Appl. Environ. Microbiol. 71: 5348-5353

**Technical comments**
Line 322: DSMZ, Braunschweig, Germany

This has been corrected.

References:

Dimikić, M., Pušić, M., Majkić-Dursun, B. & Obradović, V. Certain implications of oxic conditions in alluvial groundwater. *Water Res. Manage.* **1(2)**, 27-43, (2011).

Fike, D.A., Gammon, C.L., Ziebis, W. & Orphan, V.J. Micron-scale mapping of sulfur cycling across the oxycline of a cyanobacterial mat: a paired nanoSIMS and CARD-FISH approach. *ISME J.* **2**, 749-759, (2008).

Kato, M.T., Field, J.A. & Lettinga, G. Anaerobe tolerance to oxygen and the potentials of anaerobic and aerobic cocultures for wastewater treatment. *Braz. J. Chem. Eng.* **14(4)**, (1997).

Kjeldsen, K.U., Joulian, C. & Ingvorsen, K. Oxygen tolerance of sulfate-reducing bacteria in activated sludge. *Environ. Sci. Technol.* **38(7)**, 2038-2043, (2004).

Maamar, S.B., Aquilina, L., Quaiser, A., Pauwels, H., Michon-Coudouel, S., Vergnaud-Ayraud, V., Labasque, T., Roques, C., Abbott, B.W. & Dufresne, A. Groundwater Isolation Governs Chemistry and Microbial Community Structure along Hydrologic Flowpaths. *Fron. Microbiol.* **6**: 1457, (2015).

Pester, M., Schleper, C., Wagner, M. (2011) The Thaumarchaeota: an emerging view of their phylogeny and ecophysiology. Current Opinion in Microbiology, 14: 300-306.

---

## Author Comment (AC2) · 25 Nov 2016

We thank the reviewer for their time and constructive comments on our manuscript. We have addressed all concerns raised below.

**M. Currell (Referee)**

Iverach et al. present a novel approach to the determination of methane sources in shallow groundwater in the Condamine Alluvium aquifer, Australia. I think the study is of high scientific significance, for two main reasons:

1. The use of combined geochemical and microbiological indicators to study the origins of methane in groundwater is novel. Studies of this kind are relatively rare in the literature, and the microbiological analysis provide insight about the methane sources and degradation processes that couldn't otherwise be gained from the isotopic analyses alone
2. The topic and research question(s) are of high importance, given the current debate about environmental impacts of coal seam gas (and other unconventional gas), both in this particular area of Australia, and worldwide.

There are some minor issues and corrections needed, and some areas where additional information could be included to make the paper more solid. However, overall I think this is a high quality manuscript.

Specific comments

**Abstract Line 33-34**: Which data? I like to see some actual data values or description of the particular aspects of the data set of greatest significance (and supporting the conclusions described) included in the abstract. If more space is needed in order to do this, I suggest removing the second sentence of the abstract, as this is background information that can be included in the introduction.
A description of the particular data that provide the greatest significance (no methanogenesis *in-situ*) has been included in the abstract. We mention the isotopes of DIC and DOC and the concentration of $SO_4^{2-}$ as being the pertinent geochemical data, and the absence of methanogenic archaea being the important microbial data presented to support the conclusions in the manuscript.

**Introduction Line 50**: I suggest adding the term 'in situ' when discussing biological production of methane in the shallow groundwater. This makes it clear that you are distinguishing two different potential gas sources- one produced in the shallow aquifer itself, and another whereby gas from another unit has migrated to the aquifer.
We have added 'in situ' when discussing biological production of methane in the shallow groundwater.

**Line 81:** 'Therefore' is not really the best word here. It does not follow logically from the preceding discussion that combining geochemistry/microbiology can discriminate the relevant processes; rather you could say that microbiological indicators have the potential to resolve some of the uncertainties just mentioned (e.g. methanogenesis and methane degradation processes), that can't be otherwise determined on the basis of geochemical data alone. Here you could also note the general absence of published studies which have combined geochemical and microbiological indicators to look at methane sources and degradation in an applied setting (an important point to make in your introduction).
'Therefore' has been removed and sentence has been rewritten following the suggestion

above. We have also mentioned that there are no studies using geochemical and microbiological indicators to assess $CH_4$ production and degradation processes in a freshwater aquifer and that this study aims to fill this gap in the literature.

**Line 103:** See previous comment; this could be clarified by adding 'in situ methanogenesis' to distinguish from gas migration from another unit.
As above, the term 'in situ' has now been added.

**Line 104-108:** I think you should expand this paragraph and include some of the actual data, e.g. the observed ranges and mean/median values of d13CCH4 and d13CDIC found in the WCM from other published studies. This can be included in the text (e.g. ranges, mean values etc), as well as in a table. This would help to strengthen your isotopic lines of evidence to support the hypothesised migration mechanism later in the manuscript. Note that Baublys et al 2015 (Int. J. Coal Geol v.147-8, pp85-104) have also reported extensive data on isotopic composition of gases and water in the WCM, which should be included along with other recent published studies.
This paragraph has been expanded to include some actual data reported for the WCM. Data ranges have been provided in text as well as in a table. Isotopes for DIC weren't available for all of the studies, but included where possible. Baublys et al. 2015 has been added to the references here.

**Study area Line 146-47**: Try to avoid repetition (primary/primarily)
Primarily has been removed from the second sentence.

**Line 151**: Suggest adding 'including methane concentrations' at the end of this sentence, to highlight the significance of what you are looking at (mostly the methane in groundwater).
We have added 'especially with respect to $CH_4$ concentrations' after groundwater quality to highlight that it is the methane in groundwater that we are concerned with.

**2.1 Hydrogeological setting**. Could you include a cross section or at least a stratigraphic column to go with your description of the geological units?
We have included a cross section to go with the description and provided a reference to the literature.

**Line 160:** 'The CRAA sits within the Surat Basin, which is a major sub-province of the Great Artesian Basin'. Perhaps refer to one of the Geoscience Australia and/or CSIRO hydrogeology reports on the GAB (e.g. Ransley and Smerdon, 2012).
We have now referenced the abovementioned report, as well as the extensive work by Radke et al. 2000 on the hydrodynamics and hydrochemistry of the GAB (Radke et al. Hydrochemistry and implied hydrodynamics of the Cadna-owie Hooray Aquifer Great Artesian Basin, 2000).

**Line 188**: The recent studies by the Office of Groundwater Impact Assessment (OGIA) may have more detail about the connectivity between the CRAA and the WCM and the extent of the aquitard(s), e.g. the Surat Underground water impact report (OGIA, 2016).
This reference has been included in the connectivity section of the hydrogeology, with a sentence explaining their more recent findings on the connectivity between the WCM and the CRAA (lines 229-233).

**Line 203**: Connectivity for gas? water? both?

This has been clarified in the manuscript. It is connectivity for both gas and water.

**Method Line 212**: Here you should refer to a figure and/or table which includes your sample depths and locations

Figure 1 has been referred to in the methods for the locations of the samples and a small table has now been included to show the slotted interval depth of each bore that was sampled.

**Line 233-34**: Were the physico-chemical parameters (EC, pH, DO) monitored during the second round of sampling? If so, you could report these and use as evidence that the water composition between the two sampling events did not change substantially (if this is true).

Unfortunately, the physico-chemical parameters were not monitored during the second round of sampling. However, thirty years of studies have shown that the groundwater chemistry has remained fairly consistent (Huxley 1982).

**Line 238-239**: What about cations?

Our groundwater samples were analysed for cations, however we don't use any cation data in this manuscript. For completeness, we have now added the analysis method that the groundwater underwent for cations.

**Line 242-243**: Can you refer to a published paper where the same method was used? Same for the DIC isotopes (line 245).

Published papers have now been referred to for all of the analytical techniques used for the geochemical analyses.

**Results & Discussion Line 371**: Suggest writing 'in situ within the CRAA' instead of 'locally' to be clearer.

This has been changed.

**Line 377:** Do you mean the major ion data? Which particular aspects (e.g. sulfate and nitrate concentration data)?

At this point the discussion is just on the DIC and DOC isotopic data. The beginning of the sentence has been changed to "Our isotopic geochemical data" to make it clearer.

**Line 396**: Suggest changing to: 'major processes resulting in CH4 in the CRAA' rather than 'producing CH4 in the CRAA' (or you could say 'responsible for the presence of CH4').

This has been changed.

**Line 398**: Suggest changing 'coming from' to 'derived from'.

This has been changed.

**Line 406-411**: This paragraph is a bit confusing and needs re-writing. Is the gas in the WCM really 'typically thermogenic'? All of the isotopic data for 13CCH4 I have seen for gases and water in the WCM indicates a bacterial source of methane (e.g. 13CCH4 values around -50permil) rather than thermogenic (which should have values higher than -40permil). Is there anything else distinctive about the samples with more depleted 13CCH4, such as a much lower CH4 concentrations or differences in the major ions that could explain the isotopic difference?

Stating that the gas from the WCM was thermogenic was a large oversight and this sentence has now actually been completely removed in the re-write of the paragraph. A new reference that was published after this manuscript was originally submitted has been added (Owen et

al., 2016). This paper describes an isotopic signature for a 'shallow WCM' – a unit between the WCM 'gas reservoir' and the overlying alluvium. This signature is between -80permil and -65permil. Therefore, the -69.1permil that these three samples exhibit (despite no methanogens) could be a result of $CH_4$ from this 'shallow WCM', rather than the deeper 'gas reservoir'. This is discussed in text now.

**Line 431-432**: Yes, and further, the evidence about the presence of sulfate and conditions favouring SRB is a further line of evidence that in situ methanogenesis is unlikely to be responsible for the CH4 in the shallow aquifer
This further line of evidence has been included to strengthen the manuscript.

**Line 434 - 476**: The section on methane oxidation is insightful; good use of the microbiological methods to combine with the isotopic data and yield some new insights.
Thank you.

**Line 478**: Use the full name for AOM in the title.
The full name for AOM is now used in the title

**Line 499-500:** Relative to what? Other water in the CRAA?
It was relative to groundwaters that have the potential for AOM to occur via denitrification. This has been clarified in text with the appropriate reference.

**General comment** I think including a figure showing your isotopic compositions (13CCH4) and concentrations of methane, (using the data from Iverach 2015) and comparing with other published data on isotopic characteristics of WCM gases would be helpful, to strengthen the evidence for the proposed hypothesis (together with the microbiological indicators).
A conceptual figure has been included that highlights that there is no *in situ* $CH_4$ production in the aquifer, there is the presence of $CH_4$ in the aquifer and there are abundant $CH_4$ oxidisers in the aquifer. Hence, there is $CH_4$ migrating upwards to provide the substrate for those oxidisers. Isotopic signatures from the literature provided for the WCM, as well as the signature for the more depleted shallow WCM and measured isotopic signatures for the CRAA (from Iverach et al. 2015) have been included.

[Figure]

**Conclusions Line 536**: You could also note your other lines of evidence here (e.g., that this is supported by the co-existence of CH4 with sulfate in the groundwater, and the isotopic

composition of the methane).

The isotopic signature of $CH_4$ and the concentration of $SO_4$ have been added as further evidence (on top of the microbial data) that methane is being oxidised (hence needs a source to oxidise) and is not being produced *in-situ.*

**Line 547-548:** Your study does not really provide information about the precise pathway(s) by which methane migrates from the WCM to the CRAA, only strong evidence that such migration occurs. Hence, the statement about 'through natural faults and fractures' is really just speculation. Unless you can support it with some geological evidence, other mechanisms may also be responsible (such as transport along wells that are not fully sealed, direct leakage of gas between the units where the aquitard is absent). I suggest either talking about all possible path ways (including these), or simply leaving out the discussion of the pathway altogether and sticking to what your data shows.

We have included all pathways that the gas could be taking to migrate upwards.

---

## Author Comment (AC3) · 25 Nov 2016

Dear D.D.R. Owen,

We thank you for your detailed comments on our manuscript and have endeavoured to address your concerns with this reply. All discussion pertaining to Iverach et al. (2015) (unless pertinent to the current manuscript) has not been addressed, as that paper is not in review here. We will address your concerns raised about that manuscript in a future publication as it is a very detailed discussion.

Our sections below align with your point listing.

Section 1. "Addressing a recent paper published in the study area"

We thank you for alerting us to the publication of Owen et al. (2016), published after our submission, and we will certainly reference this work in future iterations of the manuscript.

With regard to "I also encourage the authors to contribute to the conceptual understanding of the system which could be built upon", we believe that our manuscript is making a considerable advance to the conceptualisation of processes occurring in the Condamine Alluvium. Here we provide for the first time microbiological data, which indicate that where $CH_4$ is detected in the alluvial groundwater it is likely to be sourced from the underlying geological formations.

With respect to your concerns on the usage of isotopic signature, we use the term "signature" in the same context as hundreds of other publications, most notably Whiticar (1999).

Section 2. "Thermogenic vs biogenic $CH_4$"

In portions of the manuscript we loosely used the term thermogenic when discussing $CH_4$ within the WCM (and this was also noted by our first reviewer). We will correct this to be in line with the literature. However, this does not change the interpretations in the manuscript.

Section 3. "Limitations on using $\delta^{13}$C-$CH_4$ as 'signatures'"

We are aware of the limitations of using $\delta^{13}$C-$CH_4$ to attribute source. Please see our manuscript Zazzeri et al. (2016) doi:10.5194/acp-2016-235, and papers cited in that manuscript.
http://www.atmos-chem-phys-discuss.net/acp-2016-235/. Nonetheless, $CH_4$ is a useful tracer when interpreted in the context of other hydrogeochemical and microbiological data.

The conclusions reached in this manuscript do not rely on knowledge of the isotopic values of $CH_4$. There is $CH_4$ in the alluvium, but our microbiological analyses did not identify any known microbes in the water extracted from the alluvium that would produce $CH_4$, hence it is sourced elsewhere. We propose in this manuscript that it is sourced from the underlying coal measures. We will leave our isotopic data in the revised manuscripts, because it is useful for readers of the manuscript to have knowledge of these data.

Section 4. "Identifying the $\delta^{13}$C-$CH_4$ coal measure end-member"

As stated above, identifying the $\delta^{13}$C-$CH_4$ end-member of the coal, whilst extremely useful, is not critical for assessing the microbiological processes active in the alluvium characterised

in this manuscript for the first time. In our revised manuscript we will refine the papers cited about $CH_4$ data from the WCM, including referencing Owen et al. (2016).

Section 5. "Microbial activity in the alluvium"

Thank you for drawing our attention to the lack of detail in our sampling protocol. Samples were not filtered in the field, otherwise (as you correctly state) all the biomass would have been left behind on the filter. A complete 16s rRNA sequencing was carried out on the groundwater, fully characterising the bacterial and archaeal gene targets and functional gene targets. No methanogenic archaea were ignored. Culturing experiments would be beneficial for future studies, however they were outside the scope of this research. Microbial data are useful because they tell us what is active within the alluvium. Therefore, if we have $CH_4$ but no methanogens, or oxidised $CH_4$ but no methanotrophs, this suggests that the $CH_4$ is sourced outside of the alluvium and transported upwards to the point of measurement.

As you state, the aquifer is very heterogeneous. We would like to highlight that neither in this manuscript nor in Iverach et al. (2015) did we state that the WCM had a homogeneous $\delta^{13}C$-$CH_4$ value.

We use the isotopes of DIC and DOC to help us understand what might be occurring in the aquifer. Our hard microbiological data show us that there are no methanogens present, hence there is no methanogenesis occurring in the aquifer. The isotopes of DIC and DOC have simply been employed as a secondary source of data. The fact that they show a trend indicative of no methanogenesis is completely expected, given the absence of methanogens. As you correctly state, the aquifer is heterogeneous and the samples have been taken at different depths, over large intervals – but this just makes it even more interesting that at no location in the aquifer are methanogens found.

You state that an inverse relationship between $SO_4^{2-}$ and $CH_4$ is typical of methanogenic activity (Owen et al. 2016). However, this does not prove that there is methanogenic activity in the alluvial aquifer. Here we present hard microbiological data that do not support the speculations in Owen et al. (2016).

Section 6. "$\delta^{13}C$-$CH_4$ are similar to background atmospheric $CH_4$"

Most of this section is a review of Iverach et al. (2015). Reviewing a published, peer-reviewed paper is not the focus of this discussion forum. We highlight again that the isotopic $CH_4$ data are not core to the key findings presented in this manuscript. The novel insights into the microbial communities active in the water extracted from the alluvium do not depend upon knowledge of the isotopic composition of the $CH_4$.

Section 7. Inferred vs. measured $\delta^{13}C$-$CH_4$ values.

We will modify our wording to stress the difference between measured values and inferred source values.

Section 8. "No samples were taken from the allegedly discharging aquifer".

The focus of our research is on understanding the microbiological communities present in the alluvium. It is not a study on the underlying WCM. Owen et al. (2016) now provide useful

data on the strata underlying the alluvium presenting $CH_4$ data from both the WCM and the alluvium. Here we show that there are no known methanogens in the alluvium that could produce the $CH_4$ detected in the alluvial groundwater for the data presented in either Owen et al. (2016) or Iverach et al (2015). It is therefore reasonable to propose that the $CH_4$ is sourced from the underlying rock strata (including the WCM).

Please note that our discussion focuses on the migration of $CH_4$, not the movement of water between the Walloon Coal Measures and the Condamine Alluvium.

Section 9. "Incorrect referencing"

Both Kelly and Merrick (2007) and Hillier (2010) provide useful background information to inform the discussions on hydrogeological processes within the Condamine Alluvium. Kelly and Merrick (2007) was in fact peer-reviewed by scientists from various government departments. This report was written for the Cotton Catchment Communities Cooperative Research Centre. The complete series of groundwater knowledge and gaps documents were peer-reviewed at the time of first publication and then again as part of an independent audit. Owen and Cox (2015) reference both these articles.

With respect to Kelly et al. (2014) and Duvert et al. (2015), we will replace these references with Huxley (1982) and Dafny and Silburn (2014).

In regard to "Owen and Cox (2015) did not explicitly assess water movement between the two aquifers" we provide the following quotes from that paper:

*"4.1.1. Potential connectivity between the alluvium and CSG groundwater (Cluster A)"*

*"A number of simple mixing scenarios were performed to test the likelihood that the hydrochemistry of 2005 water sample from well 42231169 was due to the influx of CSG groundwater from the WCM"*

*"It also indicates that B4 water samples are not indicative of mixing with the underlying WCM groundwater that is typically Na–HCO3–Cl water type, and B4 water types were not observed in bedrock underlying the alluvium in this area."*

*"In general, no relationships were observed between CSG groundwater in the WCM and the alluvial groundwater."*

*"as a result recharge processes and alluvium–bedrock connectivity were a focus of this study"*

Kind regards,

C.P. Iverach,
on behalf of all authors.